# Variance Reduction for Matrix Games

**Yair Carmon, Yujia Jin, Aaron Sidford and Kevin Tian**
Stanford University
{yairc,yujiajin,sidford,kjtian}@stanford.edu

## Abstract

We present a randomized primal-dual algorithm that solves the problem $\min_x \max_y y^\top Ax$ to additive error $\epsilon$ in time $\mathrm{nnz}(A) + \sqrt{\mathrm{nnz}(A)n}/\epsilon$, for matrix $A$ with larger dimension $n$ and $\mathrm{nnz}(A)$ nonzero entries. This improves the best known exact gradient methods by a factor of $\sqrt{\mathrm{nnz}(A)/n}$ and is faster than fully stochastic gradient methods in the accurate and/or sparse regime $\epsilon \leq \sqrt{n/\mathrm{nnz}(A)}$. Our results hold for $x, y$ in the simplex (matrix games, linear programming) and for $x$ in an $\ell_2$ ball and $y$ in the simplex (perceptron / SVM, minimum enclosing ball). Our algorithm combines the Nemirovski's "conceptual prox-method" and a novel reduced-variance gradient estimator based on "sampling from the difference" between the current iterate and a reference point.

## 1   Introduction

Minimax problems—or games—of the form $\min_x \max_y f(x, y)$ are ubiquitous in economics, statistics, optimization and machine learning. In recent years, minimax formulations for neural network training rose to prominence [15, 23], leading to intense interest in algorithms for solving large scale minimax games [10, 14, 20, 9, 18, 24]. However, the algorithmic toolbox for minimax optimization is not as complete as the one for minimization. Variance reduction, a technique for improving stochastic gradient estimators by introducing control variates, stands as a case in point. A multitude of variance reduction schemes exist for finite-sum minimization [cf. 19, 34, 1, 4, 12], and their impact on complexity is well-understood [43]. In contrast, only a few works apply variance reduction to finite-sum minimax problems [3, 39, 5, 26], and the potential gains from variance reduction are not well-understood.

We take a step towards closing this gap by designing variance-reduced minimax game solvers that offer strict runtime improvements over non-stochastic gradient methods, similar to that of optimal variance reduction methods for finite-sum minimization. To achieve this, we focus on the fundamental class of bilinear minimax games,

$$\min_{x \in \mathcal{X}} \max_{y \in \mathcal{Y}} y^\top Ax, \ \text{ where } \ A \in \mathbb{R}^{m \times n}.$$

In particular, we study the complexity of finding an $\epsilon$-approximate saddle point (Nash equilibrium), namely $x, y$ with

$$\max_{y' \in \mathcal{Y}} (y')^\top Ax - \min_{x' \in \mathcal{X}} y^\top Ax' \leq \epsilon.$$

In the setting where $\mathcal{X}$ and $\mathcal{Y}$ are both probability simplices, the problem corresponds to finding an approximate (mixed) equilibrium in a matrix game, a central object in game theory and economics. Matrix games are also fundamental to algorithm design due in part to their equivalence to linear programming [8]. Alternatively, when $\mathcal{X}$ is an $\ell_2$ ball and $\mathcal{Y}$ is a simplex, solving the corresponding problem finds a maximum-margin linear classifier (hard-margin SVM), a fundamental task in machine learning and statistics [25]. We refer to the former as an $\ell_1$-$\ell_1$ game and the latter as an $\ell_2$-$\ell_1$ game; our primary focus is to give improved algorithms for these domains.

## 1.1 Our Approach

Our starting point is Nemirovski's "conceptual prox-method" [28] for solving $\min_{x \in \mathcal{X}} \max_{y \in \mathcal{Y}} f(x, y)$, where $f : \mathcal{X} \times \mathcal{Y} \to \mathbb{R}$ is convex in $x$ and concave in $y$. The method solves a sequence of subproblems parameterized by $\alpha > 0$, each of the form

$$\text{find } x, y \text{ s.t. } \forall x', y' \; \langle \nabla_x f(x, y), x - x' \rangle - \langle \nabla_y f(x, y), y - y' \rangle \leq \alpha V_{x_0}(x') + \alpha V_{y_0}(y') \quad (1)$$

for some $(x_0, y_0) \in \mathcal{X} \times \mathcal{Y}$, where $V_a(b)$ is a norm-suitable Bregman divergence from $a$ to $b$: squared Euclidean distance for $\ell_2$ and KL divergence for $\ell_1$. Combining each subproblem solution with an extragradient step, the prox-method solves the original problem to $\epsilon$ accuracy by solving $\widetilde{O}(\alpha/\epsilon)$ subproblems.[1] (Solving (1) with $\alpha = 0$ is equivalent to to solving $\min_{x \in \mathcal{X}} \max_{y \in \mathcal{Y}} f(x, y)$.)

Our first contribution is showing that if a stochastic unbiased gradient estimator $\tilde{g}$ satisfies the "variance" bound

$$\mathbb{E} \|\tilde{g}(x, y) - \nabla f(x_0, y_0)\|_*^2 \leq L^2 \|x - x_0\|^2 + L^2 \|y - y_0\|^2 \quad (2)$$

for some $L > 0$, then $O(L^2/\alpha^2)$ regularized stochastic mirror descent steps using $\tilde{g}$ solve (1) in a suitable probabilistic sense. We call unbiased gradient estimators that satisfy (2) "centered."

Our second contribution is the construction of "centered" gradient estimators for $\ell_1$-$\ell_1$ and $\ell_2$-$\ell_1$ bilinear games, where $f(x, y) = y^\top A x$. Our $\ell_1$ estimator has the following form. Suppose we wish to estimate $g^{\mathsf{x}} = A^\top y$ (the gradient of $f$ w.r.t. $x$), and we already have $g_0^{\mathsf{x}} = A^\top y_0$. Let $p \in \Delta^m$ be some distribution over $\{1, \ldots, m\}$, draw $i \sim p$ and set

$$\tilde{g}^{\mathsf{x}} = g_0^{\mathsf{x}} + A_{i:} \frac{[y]_i - [y_0]_i}{p_i},$$

where $A_{i:}$ is the $i$th column of $A^\top$. This form is familiar from variance reduction techniques [19, 44, 1], that typically use a fixed distribution $p$. In our setting, however, a fixed $p$ will not produce sufficiently low variance. Departing from prior variance-reduction work and building on [16, 6], we choose $p$ *based on* $y$ according to

$$p_i(y) = \frac{\left| [y]_i - [y_0]_i \right|}{\|y - y_0\|_1},$$

yielding exactly the variance bound we require. We call this technique "sampling from the difference."

For our $\ell_2$ gradient estimator, we sample from the *squared* difference, drawing $\mathcal{X}$-block coordinate $j \sim q$, where $q_j(x) = ([x]_j - [x_0]_j)^2 / \|x - x_0\|_2^2$. To strengthen our results for $\ell_2$-$\ell_1$ games, we consider a refined version of the "centered" criterion (2) which allows regret analysis using local norms [37, 6]. To further facilitate this analysis we follow [6] and introduce gradient clipping. We extend our proofs to show that stochastic regularized mirror descent can solve (1) despite the (distance-bounded) bias caused by gradient clipping.

Our gradient estimators attain the bound (2) with $L$ equal to the Lipschitz constant of $\nabla f$. Specifically,

$$L = \begin{cases} \max_{ij} |A_{ij}| & \text{in the } \ell_1\text{-}\ell_1 \text{ setup} \\ \max_i \|A_{i:}\|_2 & \text{in the } \ell_2\text{-}\ell_1 \text{ setup.} \end{cases} \quad (3)$$

## 1.2 Method complexity compared with prior art

As per the discussion above, to achieve accuracy $\epsilon$ our algorithm solves $\widetilde{O}(\alpha/\epsilon)$ subproblems. Each subproblem takes $O(\text{nnz}(A))$ time for computing two exact gradients (one for variance reduction and one for an extragradient step), plus an additional $(m + n)L^2/\alpha^2$ time for the inner mirror descent iterations, with $L$ as in (3). The total runtime is therefore

$$\widetilde{O}\left( \left( \text{nnz}(A) + \frac{(m+n)L^2}{\alpha^2} \right) \frac{\alpha}{\epsilon} \right).$$

By setting $\alpha$ optimally to be $\max\{\epsilon, L\sqrt{(m+n)/\text{nnz}(A)}\}$, we obtain the runtime

$$\widetilde{O}(\text{nnz}(A) + \sqrt{\text{nnz}(A) \cdot (m+n)} \cdot L \cdot \epsilon^{-1}). \qquad (4)$$

**Comparison with mirror-prox and dual extrapolation.** Nemirovski [28] instantiates his conceptual prox-method by solving the relaxed proximal problem (1) with $\alpha = L$ in time $O(\text{nnz}(A))$, where $L$ is the Lipschitz constant of $\nabla f$, as given in (3). The total complexity of the resulting method is therefore

$$\widetilde{O}(\text{nnz}(A) \cdot L \cdot \epsilon^{-1}). \qquad (5)$$

The closely related dual extrapolation method of Nesterov [31] attains the same rate of convergence. We refer to the running time (5) as *linear* since it scales linearly with the problem description size $\text{nnz}(A)$. Our running time guarantee (4) is never worse than (5) by more than a constant factor, and improves on (5) when $\text{nnz}(A) = \omega(n+m)$, i.e. whenever $A$ is not extremely sparse. In that regime, our method uses $\alpha \ll L$, hence solving a harder version of (1) than possible for mirror-prox.

**Comparison with sublinear-time methods** Using a randomized algorithm, Grigoriadis and Khachiyan [16] solve $\ell_1$-$\ell_1$ bilinear games in time

$$\widetilde{O}((m+n) \cdot L^2 \cdot \epsilon^{-2}), \qquad (6)$$

and Clarkson et al. [6] extend this result to $\ell_2$-$\ell_1$ bilinear games, with the values of $L$ as in (3). Since these runtimes scale with $n+m \leq \text{nnz}(A)$, we refer to them as *sublinear*. Our guarantee improves on the guarantee (6) when $(m+n) \cdot L^2 \cdot \epsilon^{-2} \gg \text{nnz}(A)$, i.e. whenever (6) is not truly sublinear.

Our method carefully balances linear-time extragradient steps with cheap sublinear-time stochastic gradient steps. Consequently, our runtime guarantee (4) inherits strengths from both the linear and sublinear runtimes. First, our runtime scales linearly with $L/\epsilon$ rather than quadratically, as does the linear runtime (5). Second, while our runtime is not strictly sublinear, its component proportional to $L/\epsilon$ is $\sqrt{\text{nnz}(A)(n+m)}$, which is sublinear in $\text{nnz}(A)$.

Overall, our method offers the best runtime guarantee in the literature in the regime

$$\frac{\sqrt{\text{nnz}(A)(n+m)}}{\min\{n,m\}^\omega} \ll \frac{\epsilon}{L} \ll \sqrt{\frac{n+m}{\text{nnz}(A)}},$$

where the lower bound on $\epsilon$ is due to the best known theoretical runtimes of interior point methods: $\widetilde{O}(\max\{n,m\}^\omega \log(L/\epsilon))$ [7] and $\widetilde{O}(\text{nnz}(A) + \min\{n,m\}^2)\sqrt{\min\{n,m\}} \log(L/\epsilon))$ [21], where $\omega$ is the (current) matrix multiplication exponent.

In the square dense case (i.e. $\text{nnz}(A) \approx n^2 = m^2$), we improve on the accelerated runtime (5) by a factor of $\sqrt{n}$, the same improvement that optimal variance-reduced finite-sum minimization methods achieve over the fast gradient method [44, 1].

## 1.3 Related work

Matrix games, the canonical form of discrete zero-sum games, have long been studied in economics [32]. The classical mirror descent (i.e. no-regret) method yields an algorithm with running time $\widetilde{O}(\text{nnz}(A)L^2\epsilon^{-2})$ [30]. Subsequent work [16, 28, 31, 6] improve this runtime as described above. Our work builds on the extragradient scheme of Nemirovski [28] as well as the gradient estimation and clipping technique of Clarkson et al. [6].

Balamurugan and Bach [3] apply standard variance reduction [19] to bilinear $\ell_2$-$\ell_2$ games by sampling elements proportional to squared matrix entries. Using proximal-point acceleration they obtain a runtime of $\widetilde{O}(\text{nnz}(A) + \|A\|_F \sqrt{\text{nnz}(A)\max\{m,n\}}\epsilon^{-1}\log\frac{1}{\epsilon})$, a rate we recover using our algorithm (Appendix E). However, in this setting the mirror-prox method has runtime $\widetilde{O}(\|A\|_{\text{op}}\text{nnz}(A)\epsilon^{-1})$, which may be better than the result of [3] by a factor of $\sqrt{mn/\text{nnz}(A)}$ due to the discrepancy in the norm of $A$. Naive application of [3] to $\ell_1$ domains results in even greater potential losses. Shi et al. [39] extend the method of [3] to smooth functions using general Bregman divergences, but their extension is unaccelerated and appears limited to a $\epsilon^{-2}$ rate.

Chavdarova et al. [5] propose a variance-reduced extragradient method with applications to generative adversarial training. In contrast to our algorithm, which performs extragadient steps in the outer loop,

the method of [5] performs stochastic extragradient steps in the inner loop, using finite-sum variance reduction as in [19]. Chavdarova et al. [5] analyze their method in the convex-concave setting, showing improved stability over direct application of the extragradient method to noisy gradients. However, their complexity guarantees are worse than those of linear-time methods. Following up on [5], Mishchenko et al. [26] propose to reduce the variance of the stochastic extragradient method by using the same stochastic sample for both the gradient and extragradient steps. In the Euclidean strongly convex case, they show a convergence guarantee with a relaxed variance assumption, and in the noiseless full-rank bilinear case they recover the guarantees of [27]. In the general convex case, however, they only show an $\epsilon^{-2}$ rate of convergence.

## 1.4 Paper outline and additional contributions

We define our notation in Section 2. In Section 3.1, we review Nemirovski's conceptual prox-method and introduce the notion of a relaxed proximal oracle; we implement such oracle using variance-reduced gradient estimators in Section 3.2. In Section 4, we construct these gradient estimators for the $\ell_1$-$\ell_1$ and $\ell_2$-$\ell_1$ domain settings, and complete the analyses of the corresponding algorithms; in Appendix E we provide analogous treatment for the $\ell_2$-$\ell_2$ setting, recovering the results of [3].

In Appendix F we provide three additional contributions: variance-reduction-based computation of proximal points for arbitrary convex-concave functions (Appendix F.1); extension of our results to "composite" saddle point problems of the form $\min_{x \in \mathcal{X}} \max_{y \in \mathcal{Y}} \{f(x, y) + \phi(x) - \psi(y)\}$, where $f$ admits a centered gradient estimator and $\phi, \psi$ are "simple" convex functions (Appendix F.2); and a number of alternative centered gradient estimators for the $\ell_2$-$\ell_1$ and $\ell_2$-$\ell_2$ settings (Appendix F.3).

## 2 Notation

**Problem setup.** A *setup* is the triplet $(\mathcal{Z}, \|\cdot\|, r)$ where: (i) $\mathcal{Z}$ is a compact and convex subset of $\mathbb{R}^n \times \mathbb{R}^m$, (ii) $\|\cdot\|$ is a norm on $\mathcal{Z}$ and (iii) $r$ is 1-strongly-convex w.r.t. $\mathcal{Z}$ and $\|\cdot\|$, i.e. such that $r(z') \geq r(z) + \langle \nabla r(z), z - z' \rangle + \frac{1}{2} \|z' - z\|^2$ for all $z, z' \in \mathcal{Z}$.[2] We call $r$ the *distance generating function* and denote the *Bregman divergence* associated with it by

$$V_z(z') := r(z') - r(z) - \langle \nabla r(z), z' - z \rangle \geq \frac{1}{2} \|z' - z\|^2.$$

We also denote $\Theta := \max_{z'} r(z') - \min_z r(z)$ and assume it is finite.

**Norms and dual norms.** We write $\mathcal{S}^*$ for the set of linear functions on $\mathcal{S}$. For $\zeta \in \mathcal{Z}^*$ we define the dual norm of $\|\cdot\|$ as $\|\zeta\|_* := \max_{\|z\| \leq 1} \langle \zeta, z \rangle$. For $p \geq 1$ we write the $\ell_p$ norm $\|z\|_p = (\sum_i z_i^p)^{1/p}$ with $\|z\|_\infty = \max_i |z_i|$. The dual norm of $\ell_p$ is $\ell_q$ with $q^{-1} = 1 - p^{-1}$.

**Domain components.** We assume $\mathcal{Z}$ is of the form $\mathcal{X} \times \mathcal{Y}$ for convex and compact sets $\mathcal{X} \subset \mathbb{R}^n$ and $\mathcal{Y} \subset \mathbb{R}^m$. Particular sets of interest are the simplex $\Delta^d = \{v \in \mathbb{R}^d \mid \|v\|_1 = 1, v \geq 0\}$ and the Euclidean ball $\mathbb{B}^d = \{v \in \mathbb{R}^d \mid \|v\|_2 \leq 1\}$. For any vector in $z \in \mathbb{R}^n \times \mathbb{R}^m$,

we write $z^\mathsf{x}$ and $z^\mathsf{y}$ for the first $n$ and last $m$ coordinates of $z$, respectively.

When totally clear from context, we sometimes refer to the $\mathcal{X}$ and $\mathcal{Y}$ components of $z$ directly as $x$ and $y$. We write the $i$th coordinate of vector $v$ as $[v]_i$.

**Matrices.** We consider a matrix $A \in \mathbb{R}^{m \times n}$ and write $\mathrm{nnz}(A)$ for the number of its nonzero entries. For $i \in [n]$ and $j \in [m]$ we write $A_{i:}$, $A_{:j}$ and $A_{ij}$ for the corresponding row, column and entry, respectively.[3] We consider the matrix norms $\|A\|_{\max} := \max_{ij} |A_{ij}|$, $\|A\|_{p \to q} := \max_{\|x\|_p \leq 1} \|Ax\|_q$ and $\|A\|_\mathrm{F} := (\sum_{i,j} A_{ij}^2)^{1/2}$.

# 3 Primal-dual variance reduction framework

In this section, we establish a framework for solving the saddle point problem

$$\min_{x \in \mathcal{X}} \max_{y \in \mathcal{Y}} f(x, y),$$

where $f$ is convex in $x$ and concave $y$, and admits a (variance-reduced) stochastic estimator for the continuous and monotone[4] gradient mapping

$$g(z) = g(x, y) := \left( \nabla_x f(x, y), -\nabla_y f(x, y) \right).$$

Our goal is to find an $\epsilon$-approximate saddle point (Nash equilibrium), i.e. $z \in \mathcal{Z} := \mathcal{X} \times \mathcal{Y}$ such that

$$\mathrm{Gap}(z) := \max_{y' \in \mathcal{Y}} f(z^{\mathsf{x}}, y') - \min_{x' \in \mathcal{X}} f(x', z^{\mathsf{y}}) \le \epsilon. \tag{7}$$

We achieve this by generating a sequence $z_1, z_2, \ldots, z_k$ such that $\frac{1}{K} \sum_{k=1}^{K} \langle g(z_k), z_k - u \rangle \le \epsilon$ for every $u \in \mathcal{Z}$ and using the fact that

$$\mathrm{Gap}\left( \frac{1}{K} \sum_{k=1}^{K} z_k \right) \le \max_{u \in \mathcal{Z}} \frac{1}{K} \sum_{k=1}^{K} \langle g(z_k), z_k - u \rangle \tag{8}$$

due to convexity-concavity of $f$ (see proof in Appendix A.1).

In Section 3.1 we define the notion of a (randomized) *relaxed proximal oracle*, and describe how Nemirovski's mirror-prox method leverages it to solve the problem (3). In Section 3.2 we define a class of *centered* gradient estimators, whose variance is proportional to the squared distance from a reference point. Given such a centered gradient estimator, we show that a regularized stochastic mirror descent scheme constitutes a relaxed proximal oracle. For a technical reason, we limit our oracle guarantee in Section 3.2 to the bilinear case $f(x, y) = y^\top A x$, which suffices for the applications in Section 4. We lift this limitation in Appendix F.1, where we show a different oracle implementation that is valid for general convex-concave $f$, with only a logarithmic increase in complexity.

## 3.1 The mirror-prox method with a randomized oracle

Recall that we assume the space $\mathcal{Z} = \mathcal{X} \times \mathcal{Y}$ is equipped with a norm $\|\cdot\|$ and distance generating function $r : \mathcal{Z} \to \mathbb{R}$ that is 1-strongly-convex w.r.t. $\|\cdot\|$ and has range $\Theta$. We write the induced Bregman divergence as $V_z(z') = r(z') - r(z) - \langle \nabla r(z), z' - z \rangle$. We use the following fact throughout the paper: by definition, the Bregman divergence satisfies, for any $z, z', u \in \mathcal{Z}$,

$$- \langle \nabla V_z(z'), z' - u \rangle = V_z(u) - V_{z'}(u) - V_z(z'). \tag{9}$$

For any $\alpha > 0$ we define the $\alpha$-proximal mapping $\mathrm{Prox}_z^\alpha(g)$ to be the solution of the variational inequality corresponding to the strongly monotone operator $g + \alpha \nabla V_z$, i.e. the unique $z_\alpha \in \mathcal{Z}$ such that $\langle g(z_\alpha) + \alpha \nabla V_z(z_\alpha), z_\alpha - u \rangle \le 0$ for all $u \in \mathcal{Z}$ [cf. 11]. Equivalently (by (9)),

$$\mathrm{Prox}_z^\alpha(g) := \text{the unique } z_\alpha \in \mathcal{Z} \text{ s.t. } \langle g(z_\alpha), z_\alpha - u \rangle \le \alpha V_z(u) - \alpha V_{z_\alpha}(u) - \alpha V_z(z_\alpha) \ \forall u \in \mathcal{Z}. \tag{10}$$

When $V_z(z') = V_x^{\mathsf{x}}(x') + V_y^{\mathsf{y}}(y')$, $\mathrm{Prox}_z^\alpha(g)$ is also the unique solution of the saddle point problem

$$\min_{x' \in \mathcal{X}} \max_{y' \in \mathcal{Y}} \left\{ f(x', y') + \alpha V_x^{\mathsf{x}}(x') - \alpha V_y^{\mathsf{y}}(y') \right\}.$$

Consider iterations of the form $z_k = \mathrm{Prox}_{z_{k-1}}^\alpha(g)$, with $z_0 = \arg\min_z r(z)$. Averaging the definition (10) over $k$, using the bound (8) and the nonnegativity of Bregman divergences gives

$$\mathrm{Gap}\left( \frac{1}{K} \sum_{k=1}^{K} z_k \right) \le \max_{u \in \mathcal{Z}} \frac{1}{K} \sum_{k=1}^{K} \langle g(z_k), z_k - u \rangle \le \max_{u \in \mathcal{Z}} \frac{\alpha \left( V_{z_0}(u) - V_{z_K}(u) \right)}{K} \le \frac{\alpha \Theta}{K}.$$

Thus, we can find an $\epsilon$-suboptimal point in $K = \alpha \Theta / \epsilon$ exact proximal steps. However, computing $\mathrm{Prox}_z^\alpha(g)$ exactly may be as difficult as solving the original problem. Nemirovski [28] proposes a relaxation of the exact proximal mapping, which we slightly extend to include the possibility of randomization, and formalize in the following.

**Definition 1** (($\alpha, \varepsilon$)-relaxed proximal oracle). *Let $g$ be a monotone operator and $\alpha, \varepsilon > 0$. An ($\alpha, \varepsilon$)-relaxed proximal oracle for $g$ is a (possibly randomized) mapping $\mathcal{O} : \mathcal{Z} \to \mathcal{Z}$ such that $z' = \mathcal{O}(z)$ satisfies*

$$\mathbb{E}\left[\max_{u \in \mathcal{Z}} \left\{ \langle g(z'), z' - u \rangle - \alpha V_z(u) \right\}\right] \leq \varepsilon.$$

Note that $\mathcal{O}(z) = \mathrm{Prox}_z^\alpha(g)$ is an ($\alpha, 0$)-relaxed proximal oracle. Algorithm 1 describes the "conceptual prox-method" of Nemirovski [28], which recovers the error guarantee of exact proximal iterations. The $k$th iteration consists of (i) a relaxed proximal oracle call producing $z_{k-1/2} = \mathcal{O}(z_{k-1})$, and (ii) a *linearized* proximal (mirror) step where we replace $z \mapsto g(z)$ with the constant function $z \mapsto g(z_{k-1/2})$, producing $z_k = \mathrm{Prox}_{z_{k-1}}^\alpha(g(z_{k-1/2}))$. We now state the convergence guarantee for the mirror-prox method, first shown in [28] (see Appendix B.1 for a simple proof).

---

**Algorithm 1:** OuterLoop($\mathcal{O}$) (Nemirovski [28])

---

**Input**: ($\alpha, \varepsilon$)-relaxed proximal oracle $\mathcal{O}(z)$ for gradient mapping $g$, distance-generating $r$
**Parameters**: Number of iterations $K$
**Output**: Point $\bar{z}_K$ with $\mathbb{E}\,\mathrm{Gap}(\bar{z}) \leq \frac{\alpha \Theta}{K} + \varepsilon$

1  $z_0 \leftarrow \arg\min_{z \in \mathcal{Z}} r(z)$
2  **for** $k = 1, \ldots, K$ **do**
3  $\quad z_{k-1/2} \leftarrow \mathcal{O}(z_{k-1})$      $\triangleright$ We implement $\mathcal{O}(z_{k-1})$ by calling InnerLoop($z_{k-1}, \tilde{g}_{z_{k-1}}, \alpha$)
4  $\quad z_k \leftarrow \mathrm{Prox}_{z_{k-1}}^\alpha(g(z_{k-1/2})) = \arg\min_{z \in \mathcal{Z}} \left\{ \langle g(z_{k-1/2}), z \rangle + \alpha V_{z_{k-1}}(z) \right\}$
5  **return** $\bar{z}_K = \frac{1}{K} \sum_{k=1}^K z_{k-1/2}$

---

**Proposition 1** (Mirror prox convergence via oracles). *Let $\mathcal{O}$ be an ($\alpha, \varepsilon$)-relaxed proximal oracle with respect to gradient mapping $g$ and distance-generating function $r$ with range at most $\Theta$. Let $z_{1/2}, z_{3/2}, \ldots, z_{K-1/2}$ be the iterates of Algorithm 1 and let $\bar{z}_K$ be its output. Then*

$$\mathbb{E}\,\mathrm{Gap}(\bar{z}_K) \leq \mathbb{E}\max_{u \in \mathcal{Z}} \frac{1}{K} \sum_{k=1}^K \langle g(z_{k-1/2}), z_{k-1/2} - u \rangle \leq \frac{\alpha \Theta}{K} + \varepsilon.$$

### 3.2 Implementation of an ($\alpha, 0$)-relaxed proximal oracle

We now explain how to use stochastic variance-reduced gradient estimators to design an efficient ($\alpha, 0$)-relaxed proximal oracle. We begin by introducing the bias and variance properties of the estimators we require.

**Definition 2.** *Let $z_0 \in \mathcal{Z}$ and $L > 0$. A stochastic gradient estimator $\tilde{g}_{z_0} : \mathcal{Z} \to \mathcal{Z}^*$ is called ($z_0, L$)-centered for $g$ if for all $z \in \mathcal{Z}$*

> *1. $\mathbb{E}[\tilde{g}_{z_0}(z)] = g(z)$,*
>
> *2. $\mathbb{E}\|\tilde{g}_{z_0}(z) - g(z_0)\|_*^2 \leq L^2 \|z - z_0\|^2$.*

**Lemma 1.** *A ($z_0, L$)-centered estimator for $g$ satisfies $\mathbb{E}\|\tilde{g}_{z_0}(z) - g(z)\|_*^2 \leq (2L)^2 \|z - z_0\|^2$.*

*Proof.* Writing $\tilde{\delta} = \tilde{g}_{z_0}(z) - g(z_0)$, we have $\mathbb{E}\tilde{\delta} = g(z) - g(z_0)$ by the first centered estimator property. Therefore,

$$\mathbb{E}\|\tilde{g}_{z_0}(z) - g(z)\|_*^2 = \mathbb{E}\|\tilde{\delta} - \mathbb{E}\tilde{\delta}\|_*^2 \overset{(i)}{\leq} 2\mathbb{E}\|\tilde{\delta}\|_*^2 + 2\|\mathbb{E}\tilde{\delta}\|_*^2 \overset{(ii)}{\leq} 4\mathbb{E}\|\tilde{\delta}\|_*^2 \overset{(iii)}{\leq} (2L)^2 \|z - z_0\|^2,$$

where the bounds follow from $(i)$ the triangle inequality, $(ii)$ Jensen's inequality and $(iii)$ the second centered estimator property. $\qquad\square$

**Remark 1.** A gradient mapping that admits a ($z, L$)-centered gradient estimator for every $z \in \mathcal{Z}$ is $2L$-Lipschitz, since by Jensen's inequality and Lemma 1 we have for all $w \in \mathcal{Z}$

$$\|g(w) - g(z)\|_* = \|\mathbb{E}\tilde{g}_z(w) - g(z)\|_* \leq (\mathbb{E}\|\tilde{g}_z(w) - g(z)\|_*^2)^{1/2} \leq 2L \|w - z\|.$$

**Remark 2.** Definition 2 bounds the gradient variance using the distance to the reference point. Similar bounds are used in variance reduction for bilinear saddle-point problems with Euclidean norm [3], as well as for finding stationary points in smooth nonconvex finite-sum problems [2, 33, 12, 45]. However, known variance reduction methods for smooth convex finite-sum minimization require stronger bounds [cf. 1, Section 2.1].

With the variance bounds defined, we describe Algorithm 2 which (for the bilinear case) implements a relaxed proximal oracle. The algorithm is stochastic mirror descent with an additional regularization term around the initial point $w_0$. Note that we do not perform extragradient steps in this stochastic method. When combined with a centered gradient estimator, the iterates of Algorithm 2 provide the following guarantee, which is one of our key technical contributions.

---

**Algorithm 2:** $\texttt{InnerLoop}(w_0, \tilde{g}_{w_0}, \alpha)$

**Input**: Initial $w_0 \in \mathcal{Z}$, gradient estimator $\tilde{g}_{w_0}$, oracle quality $\alpha > 0$
**Parameters**: Step size $\eta$, number of iterations $T$
**Output**: Point $\bar{w}_T$ satisfying Definition 1 (for appropriate $\tilde{g}_{w_0}, \eta, T$)
1 **for** $t = 1, \ldots, T$ **do**
2 $\quad\quad w_t \leftarrow \arg\min_{w \in \mathcal{Z}} \left\{ \langle \tilde{g}_{w_0}(w_{t-1}), w \rangle + \frac{\alpha}{2} V_{w_0}(w) + \frac{1}{\eta} V_{w_{t-1}}(w) \right\}$
3 **return** $\bar{w}_T = \frac{1}{T} \sum_{t=1}^{T} w_t$

---

**Proposition 2.** *Let $\alpha, L > 0$, let $w_0 \in \mathcal{Z}$ and let $\tilde{g}_{w_0}$ be $(w_0, L)$-centered for monotone g. Then, for $\eta = \frac{\alpha}{10L^2}$ and $T \geq \frac{4}{\eta\alpha} = \frac{40L^2}{\alpha^2}$, the iterates of Algorithm 2 satisfy*

$$\mathbb{E} \max_{u \in \mathcal{Z}} \left[ \frac{1}{T} \sum_{t \in [T]} \langle g(w_t), w_t - u \rangle - \alpha V_{w_0}(u) \right] \leq 0. \tag{11}$$

Before discussing the proof of Proposition 2, we state how it implies the relaxed proximal oracle property for the bilinear case.

**Corollary 1.** *Let $A \in \mathbb{R}^{m \times n}$ and let $g(z) = (A^\top z^\mathsf{y}, -Az^\mathsf{x})$. Then, in the setting of Proposition 2, $\mathcal{O}(w_0) = \texttt{InnerLoop}(w_0, \tilde{g}_{w_0}, \alpha)$ is an $(\alpha, 0)$-relaxed proximal oracle.*

*Proof.* Note that $\langle g(z), w \rangle = -\langle g(w), z \rangle$ for any $z, w \in \mathcal{Z}$ and consequently $\langle g(z), z \rangle = 0$. Therefore, the iterates $w_1, \ldots, w_T$ of Algorithm 2 and its output $\bar{w}_T = \frac{1}{T} \sum_{t=1}^{T} w_t$ satisfy for every $u \in \mathcal{Z}$,

$$\frac{1}{T} \sum_{t \in [T]} \langle g(w_t), w_t - u \rangle = \frac{1}{T} \sum_{t \in [T]} \langle g(u), w_t \rangle = \langle g(u), \bar{w}_T \rangle = \langle g(\bar{w}_T), \bar{w}_T - u \rangle.$$

Substituting into the bound (11) yields the $(\alpha, 0)$-relaxed proximal oracle property in Definition 1. □

More generally, the proof of Corollary 1 shows that Algorithm 2 implements a relaxed proximal oracle whenever $z \mapsto \langle g(z), z - u \rangle$ is convex for every $u$. In Appendix F.1 we implement an $(\alpha, \varepsilon)$-relaxed proximal oracle without such an assumption.

The proof of Proposition 2 is a somewhat lengthy application of existing techniques for stochastic mirror descent analysis in conjunction with Definition 2. We give it in full in Appendix B.2 and sketch it briefly here. We view Algorithm 2 as mirror descent with stochastic gradients $\tilde{\delta}_t = \tilde{g}_{w_0}(w_t) - g(w_0)$ and composite term $\langle g(w_0), z \rangle + \frac{\alpha}{2} V_{w_0}(z)$. For any $u \in \mathcal{Z}$, the standard mirror descent analysis (see Lemma 4 in Appendix A.2) bounds the regret $\sum_{t \in [T]} \langle \tilde{g}_{w_0}(w_t) + \frac{\alpha}{2} \nabla V_{w_0}(w_t), w_t - u \rangle$ in terms of the distance to initialization $V_{w_0}(u)$ and the stochastic gradient norms $\|\tilde{\delta}_t\|_*^2$ for $t \in [T]$. Bounding these norms via Definition 2 and rearranging the $\langle \nabla V_{w_0}(w_t), w_t - u \rangle$ terms, we show that $\mathbb{E}\left[ \frac{1}{T} \sum_{t \in [T]} \langle g(w_t), w_t - u \rangle - \alpha V_{w_0}(u) \right] \leq 0$ for all $u \in \mathcal{Z}$. To reach our desired result we must swap the order of the expectation and "for all." We do so using the "ghost iterate" technique due to Nemirovski et al. [29].

# 4 Application to bilinear saddle point problems

We now construct centered gradient estimators (as per Definition 2) for the linear gradient mapping

$$g(z) = (A^\top z^\mathsf{y}, -Az^\mathsf{x}) \text{ corresponding to the bilinear saddle point problem } \min_{x \in \mathcal{X}} \max_{y \in \mathcal{Y}} y^\top A x.$$

Sections 4.1 and 4.2 consider the $\ell_1$-$\ell_1$ and $\ell_2$-$\ell_1$ settings, respectively; in Appendix E we show how our approach naturally extends to the $\ell_2$-$\ell_2$ setting as well. Throughout, we let $w_0$ denote the "center" (i.e. reference point) of our stochastic gradient estimator and consider a general query point $w \in \mathcal{Z} = \mathcal{X} \times \mathcal{Y}$. We also recall the notation $[v]_i$ for the $i$th entry of vector $v$.

## 4.1 $\ell_1$-$\ell_1$ games

**Setup.** Denoting the $d$-dimensional simplex by $\Delta^d$, we let $\mathcal{X} = \Delta^n$, $\mathcal{Y} = \Delta^m$ and $\mathcal{Z} = \mathcal{X} \times \mathcal{Y}$. We take $\|\cdot\|$ to be the $\ell_1$ norm with conjugate norm $\|\cdot\|_* = \|\cdot\|_\infty$. We take the distance generating function $r$ to be the negative entropy, i.e. $r(z) = \sum_i [z]_i \log[z]_i$. We note that both $\|\cdot\|_1$ and $r$ are separable and in particular separate over the $\mathcal{X}$ and $\mathcal{Y}$ blocks of $\mathcal{Z}$. Finally we set

$$\|A\|_{\max} := \max_{i,j} |A_{ij}|$$

and note that this is the Lipschitz constant of the gradient mapping $g$ under the chosen norm.

**Gradient estimator.** Given $w_0 = (w_0^\mathsf{x}, w_0^\mathsf{y})$ and $g(w_0) = (A^\top w_0^\mathsf{y}, -Aw_0^\mathsf{x})$, we describe the reduced-variance gradient estimator $\tilde{g}_{w_0}(w)$. First, we define the probabilities $p(w) \in \Delta^m$ and $q(w) \in \Delta^n$ according to,

$$p_i(w) := \frac{|[w^\mathsf{y}]_i - [w_0^\mathsf{y}]_i|}{\|w^\mathsf{y} - w_0^\mathsf{y}\|_1} \quad \text{and} \quad q_j(w) := \frac{|[w^\mathsf{x}]_j - [w_0^\mathsf{x}]_j|}{\|w^\mathsf{x} - w_0^\mathsf{x}\|_1}. \tag{12}$$

To compute $\tilde{g}_{w_0}$ we sample $i \sim p(w)$ and $j \sim q(w)$ independently, and set

$$\tilde{g}_{w_0}(w) := \left( A^\top w_0^\mathsf{y} + A_{i:} \frac{[w^\mathsf{y}]_i - [w_0^\mathsf{y}]_i}{p_i(w)}, -Aw_0^\mathsf{x} - A_{:j} \frac{[w^\mathsf{x}]_j - [w_0^\mathsf{x}]_j}{q_j(w)} \right), \tag{13}$$

where $A_{i:}$ and $A_{:j}$ are the $i$th row and $j$th column of $A$, respectively. Since the sampling distributions $p(w), q(w)$ are proportional to the absolute value of the difference between blocks of $w$ and $w_0$, we call strategy (12) "sampling from the difference." Substituting (12) into (13) gives the explicit form

$$\tilde{g}_{w_0}(w) = g(w_0) + (A_{i:}\|w^\mathsf{y} - w_0^\mathsf{y}\|_1 \text{sign}([w^\mathsf{y} - w_0^\mathsf{y}]_i), -A_{:j}\|w^\mathsf{x} - w_0^\mathsf{x}\|_1 \text{sign}([w^\mathsf{x} - w_0^\mathsf{x}]_j)). \tag{14}$$

A straightforward calculation shows that this construction satisfies Definition 2.

**Lemma 2.** *In the $\ell_1$-$\ell_1$ setup, the estimator* (14) *is* $(w_0, L)$-*centered with* $L = \|A\|_{\max}$.

*Proof.* The first property ($\mathbb{E}\tilde{g}_{w_0}(w) = g(w)$) follows immediately by inspection of (13). The second property follows from (14) by noting that

$$\|\tilde{g}_{w_0}(w) - g(w_0)\|_\infty = \max\left\{ \|A_{i:}\|_\infty \|w^\mathsf{y} - w_0^\mathsf{y}\|_1, \|A_{:j}\|_\infty \|w^\mathsf{x} - w_0^\mathsf{x}\|_1 \right\} \leq \|A\|_{\max} \|w - w_0\|_1$$

for all $i, j$, and therefore $\mathbb{E} \|\tilde{g}_{w_0}(w) - g(w_0)\|_\infty^2 \leq \|A\|_{\max}^2 \|w - w_0\|_1^2$. $\qquad \square$

The proof of Lemma 2 reveals that the proposed estimator satisfies a stronger version of Definition 2: the last property and also Lemma 1 hold with probability 1 rather than in expectation.

**Runtime bound.** Combining the centered gradient estimator (13), the relaxed oracle implementation (Algorithm 2) and the extragradient outer loop (Algorithm 1), we obtain our main result for $\ell_1$-$\ell_1$ games: an accelerated stochastic variance reduction algorithm. We write the resulting complete method explicitly as Algorithm 3 in Appendix C.1. The algorithm enjoys the following runtime guarantee (see proof in Appendix C.2).

**Theorem 1.** *Let $A \in \mathbb{R}^{m \times n}$, $\epsilon > 0$, and $\alpha \geq \epsilon / \log(nm)$. Algorithm 3 outputs a point $z = (z^{\times}, z^{\mathsf{y}})$ such that $\mathbb{E}\left[ \max_{y \in \Delta^m} y^\top A z^{\times} - \min_{x \in \Delta^n} (z^{\mathsf{y}})^\top A x \right] = \mathbb{E}\left[ \max_i [Az^{\times}]_i - \min_j [A^\top z^{\mathsf{y}}]_j \right] \leq \epsilon$, and runs in time*

$$O\left( \left( \mathrm{nnz}(A) + \frac{(m+n)\,\|A\|_{\max}^2}{\alpha^2} \right) \frac{\alpha \log(mn)}{\epsilon} \right). \tag{15}$$

*Setting $\alpha$ optimally, the running time is*

$$O\left( \mathrm{nnz}(A) + \frac{\sqrt{\mathrm{nnz}(A)(m+n)}\,\|A\|_{\max} \log(mn)}{\epsilon} \right). \tag{16}$$

### 4.2 $\ell_2$-$\ell_1$ games

**Setup.** We set $\mathcal{X} = \mathbb{B}^n$ to be the $n$-dimensional Euclidean ball of radius 1, while $\mathcal{Y} = \Delta^m$ remains the simplex. For $z = (z^{\times}, z^{\mathsf{y}}) \in \mathcal{Z} = \mathcal{X} \times \mathcal{Y}$ we define a norm by

$$\|z\|^2 = \|z^{\times}\|_2^2 + \|z^{\mathsf{y}}\|_1^2 \quad \text{with dual norm} \quad \|g\|_*^2 = \|g^{\times}\|_2^2 + \|g^{\mathsf{y}}\|_\infty^2.$$

For distance generating function we take $r(z) = r^{\times}(z^{\times}) + r^{\mathsf{y}}(z^{\mathsf{y}})$ with $r^{\times}(x) = \frac{1}{2}\|x\|_2^2$ and $r^{\mathsf{y}}(y) = \sum_i y_i \log y_i$; $r$ is 1-strongly convex w.r.t. to $\|\cdot\|$ and has range $\frac{1}{2} + \log m \leq \log(2m)$. Finally, we denote

$$\|A\|_{2\to\infty} = \max_{i \in [m]} \|A_{i:}\|_2,$$

and note that this is the Lipschitz constant of $g$ under $\|\cdot\|$.

**Gradient estimator.** To account for the fact that $\mathcal{X}$ is now the $\ell_2$ unit ball, we modify the sampling distribution $q$ in (12) to $q_j(w) = \frac{([w^{\times}]_j - [w_0^{\times}]_j)^2}{\|w^{\times} - w_0^{\times}\|_2^2}$, and keep $p$ the same. As we explain in detail in Appendix D.1.1, substituting these probabilities into the expression (13) yields a centered gradient estimator with a constant $(\sum_{j \in [n]} \|A_{:j}\|_\infty^2)^{1/2}$ that is larger than $\|A\|_{2\to\infty}$ by a factor of up to $\sqrt{n}$. Using local norms analysis allows us to tighten these bounds whenever the stochastic steps have bounded infinity norm. Following Clarkson et al. [6], we enforce such bound on the step norms via gradient clipping. The final gradient estimator is

$$\tilde{g}_{w_0}(w) := \left( A^\top w_0^{\mathsf{y}} + A_{i:} \frac{\|w^{\mathsf{y}} - w_0^{\mathsf{y}}\|_1}{\mathrm{sign}([w^{\mathsf{y}} - w_0^{\mathsf{y}}]_i)}, -A w_0^{\times} - \mathsf{T}_\tau \left( A_{:j} \frac{\|w^{\times} - w_0^{\times}\|_2^2}{[w^{\times}]_j - [w_0^{\times}]_j} \right) \right),$$

$$\text{where } [\mathsf{T}_\tau(v)]_i = \begin{cases} -\tau & [v]_i < -\tau \\ [v]_i & -\tau \leq [v]_i \leq \tau \\ \tau & [v]_i > \tau, \end{cases}$$

The clipping operation $\mathsf{T}_\tau$ introduces bias to the gradient estimator, which we account for by carefully choosing a value of $\tau$ for which the bias is on the same order as the variance, and yet the resulting steps are appropriately bounded; see Appendix D.1.2. In Appendix F.3.1 we describe an alternative gradient estimator for which the distribution $q$ does not depend on the current iterate $w$.

**Runtime bound.** Algorithm 4 in Appendix D.5 combines our clipped gradient estimator with our general variance reduction framework. The analysis in Appendix D gives the following guarantee.

**Theorem 2.** *Let $A \in \mathbb{R}^{m \times n}$, $\epsilon > 0$, and any $\alpha \geq \epsilon / \log(2m)$. Algorithm 4 outputs a point $z = (z^{\times}, z^{\mathsf{y}})$ such that $\mathbb{E}\left[ \max_{y \in \Delta^m} y^\top A z^{\times} - \min_{x \in \mathbb{B}^n} (z^{\mathsf{y}})^\top A x \right] = \mathbb{E}\left[ \max_i [Az^{\times}]_i + \|A^\top z^{\mathsf{y}}\|_2 \right] \leq \epsilon$, and runs in time*

$$O\left( \left( \mathrm{nnz}(A) + \frac{(m+n)\,\|A\|_{2\to\infty}^2}{\alpha^2} \right) \frac{\alpha \log(2m)}{\epsilon} \right). \tag{17}$$

*Setting $\alpha$ optimally, the running time is*

$$O\left( \mathrm{nnz}(A) + \frac{\sqrt{\mathrm{nnz}(A)(m+n)}\,\|A\|_{2\to\infty} \log(2m)}{\epsilon} \right). \tag{18}$$

**Acknowledgments**

YC and YJ were supported by Stanford Graduate Fellowships. AS was supported by the NSF CAREER Award CCF-1844855. KT was supported by the NSF Graduate Fellowship DGE1656518.

## Footnotes

[1] More precisely, the required number of subproblem solutions is at most $\Theta \cdot \frac{\alpha}{\epsilon}$, where $\Theta$ is a "domain size" parameter that depends on $\mathcal{X}$, $\mathcal{Y}$, and the Bregman divergence $V$ (see Section 2). In the $\ell_1$ and $\ell_2$ settings considered in this paper, we have the bound $\Theta \leq \log(nm)$ and we use the $\widetilde{O}$ notation to suppress terms logarithmic in $n$ and $m$. However, in other settings—e.g., $\ell_\infty$-$\ell_1$ games [cf. 38, 40]—making the parameter $\Theta$ scale logarithmically with the problem dimension is far more difficult.

[2] For non-differentiable $r$, let $\langle \nabla r(z), w \rangle := \sup_{\gamma \in \partial r(z)} \langle \gamma, w \rangle$, where $\partial r(z)$ is the subdifferential of $r$ at $z$.

[3] For $k \in \mathbb{N}$, we let $[k] := \{1, \dots, k\}$.

[4] A mapping $q : \mathcal{Z} \to \mathcal{Z}^*$ is monotone if and only if $\langle q(z') - q(z), z' - z \rangle \ge 0$ for all $z, z' \in \mathcal{Z}$; $g$ is monotone due to convexity-concavity of $f$.

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
