[Supplementary Material]

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

# Appendix

## A  Standard results

Below we give two standard results in convex optimization: bounding suboptimality via regret (Section A.1) and the mirror descent regret bound (Section A.2).

### A.1  Duality gap bound

Let $f : \mathcal{X} \times \mathcal{Y} \to \mathbb{R}$ be convex in $\mathcal{X}$, concave in $\mathcal{Y}$ and differentiable, and let $g(z) = g(x,y) = (\nabla_x f(x,y), -\nabla_y f(x,y))$. For $z, u \in \mathcal{Z}$ define

$$\text{gap}(z; u) := f(z^{\mathsf{x}}, u^{\mathsf{y}}) - f(u^{\mathsf{x}}, z^{\mathsf{y}}) \text{ and } \text{Gap}(z) := \max_{u \in \mathcal{Z}} \text{gap}(z; u).$$

**Lemma 3.** *For every $z_1, \ldots, z_K \in \mathcal{Z}$,*

$$\text{Gap}\left(\frac{1}{K}\sum_{k=1}^{K} z_k\right) \leq \max_{u \in \mathcal{Z}} \frac{1}{K}\sum_{k=1}^{K} \langle g(z_k), z_k - u \rangle.$$

*Proof.* Note that $\text{gap}(z; u)$ is concave in $u$ for every $z$, and that $\text{gap}(z; z) = 0$, therefore

$$\text{gap}(z; u) \leq \langle \nabla_u \text{gap}(z; z), u - z \rangle = \langle g(z), z - u \rangle.$$

Moreover, $\text{gap}(z; u)$ is convex in $z$ for every $u$. Therefore, for a sequence $z_1, \ldots, z_K$ and any $u \in \mathcal{Z}$

$$\text{gap}\left(\frac{1}{K}\sum_{k=1}^{K} z_k; u\right) \leq \frac{1}{K}\sum_{k=1}^{K} \text{gap}(z_k; u) \leq \frac{1}{K}\sum_{k=1}^{K} \langle g(z_k), z_k - u \rangle.$$

Maximizing the inequality over $u$ yields the lemma.  $\square$

### A.2  The mirror descent regret bound

Recall that $V_z(z') = r(z') - r(z) - \langle \nabla r(z), z' - z \rangle$ is the Bregman divergence induced by a 1-strongly-convex distance generating function $r$.

**Lemma 4.** *Let $Q : \mathcal{Z} \to \mathbb{R}$ be convex, let $T \in \mathbb{N}$ and let $w_0 \in \mathcal{Z}$, $\gamma_0, \gamma_1, \ldots, \gamma_T \in \mathcal{Z}^*$. The sequence $w_1, \ldots, w_T$ defined by*

$$w_t = \arg\min_{w \in \mathcal{Z}} \left\{ \langle \gamma_{t-1}, w \rangle + Q(w) + V_{w_{t-1}}(w) \right\}$$

*satisfies for all $u \in \mathcal{Z}$ (denoting $w_{T+1} := u$),*

$$\sum_{t=1}^{T} \langle \gamma_t + \nabla Q(w_t), w_t - u \rangle \leq V_{w_0}(u) + \sum_{t=0}^{T} \left\{ \langle \gamma_t, w_t - w_{t+1} \rangle - V_{w_t}(w_{t+1}) \right\}$$

$$\leq V_{w_0}(u) + \frac{1}{2}\sum_{t=0}^{T} \|\gamma_t\|_*^2.$$

*Proof.* Fix $u \equiv w_{T+1} \in \mathcal{Z}$. We note that by definition $w_t$ is the solution of a convex optimization problem with (sub)gradient $\gamma_{t-1} + \nabla Q(\cdot) + \nabla V_{w_{t-1}}(\cdot)$, and therefore by by the first-order optimality condition [cf. 17, Chapter VII] satisfies

$$\langle \gamma_{t-1} + \nabla Q(w_t) + \nabla V_{w_{t-1}}(w_t), w_t - w_{T+1} \rangle \leq 0.$$

By the equality (9) we have $-\langle \nabla V_{w_{t-1}}(w_t), w_t - w_{T+1} \rangle = V_{w_{t-1}}(w_{T+1}) - V_{w_t}(w_{T+1}) - V_{w_{t-1}}(w_t)$. Substituting and summing over $t \in [T]$ gives

$$\sum_{t=1}^{T} \langle \gamma_{t-1} + \nabla Q(w_t), w_t - w_{T+1} \rangle \leq V_{w_0}(w_{T+1}) - \sum_{t=0}^{T} V_{w_t}(w_{t+1}).$$

Rearranging the LHS and adding $\langle \gamma_T, w_T - w_{T+1} \rangle$ to both sides of the inequality gives

$$\sum_{t=1}^{T} \langle \gamma_t + \nabla Q(w_t), w_t - w_{T+1} \rangle \leq V_{w_0}(w_{T+1}) + \sum_{t=0}^{T} \left\{ \langle \gamma_t, w_t - w_{t+1} \rangle - V_{w_t}(w_{t+1}) \right\},$$

which is the first bound stated in the lemma. The second bound follows since for every $t$ we have

$$\langle \gamma_t, w_t - w_{t+1} \rangle \overset{(i)}{\leq} \|\gamma_t\|_* \|w_t - w_{t+1}\| \overset{(ii)}{\leq} \frac{1}{2} \|\gamma_t\|_*^2 + \frac{1}{2} \|w_t - w_{t+1}\|^2 \overset{(iii)}{\leq} \frac{1}{2} \|\gamma_t\|_*^2 + V_{w_t}(w_{t+1}) \tag{19}$$

due to $(i)$ Hölder's inquality, $(ii)$ Young's inequality and $(iii)$ strong convexity of $r$. $\qquad\square$

## B  Proofs from Section 3

### B.1  Derivation of the Nemirovski's conceptual prox-method

**Proposition 1** (Mirror prox convergence via oracles). *Let $\mathcal{O}$ be an $(\alpha,\varepsilon)$-relaxed proximal oracle with respect to gradient mapping $g$ and distance-generating function $r$ with range at most $\Theta$. Let $z_{1/2}, z_{3/2}, \ldots, z_{K-1/2}$ be the iterates of Algorithm 1 and let $\bar{z}_K$ be its output. Then*

$$\mathbb{E}\,\mathrm{Gap}(\bar{z}_K) \leq \mathbb{E} \max_{u \in \mathcal{Z}} \frac{1}{K} \sum_{k=1}^{K} \left\langle g(z_{k-1/2}), z_{k-1/2} - u \right\rangle \leq \frac{\alpha \Theta}{K} + \varepsilon.$$

*Proof.* Fix iteration $k$, and note that by the definition (10), $z_k = \mathrm{Prox}_{z_{k-1}}^{\alpha}(g(z_{k-1/2}))$ satisfies

$$\left\langle g(z_{k-1/2}), z_k - u \right\rangle \leq \alpha \left( V_{z_{k-1}}(u) - V_{z_k}(u) - V_{z_{k-1}}(z_k) \right) \quad \forall u \in \mathcal{Z}.$$

Summing over $k$, writing

$$\left\langle g(z_{k-1/2}), z_k - u \right\rangle = \left\langle g(z_{k-1/2}), z_{k-1/2} - u \right\rangle - \left\langle g(z_{k-1/2}), z_{k-1/2} - z_k \right\rangle$$

and rearranging yields

$$\sum_{k=1}^{K} \left\langle g(z_{k-1/2}), z_{k-1/2} - u \right\rangle \leq \alpha V_{z_0}(u) + \sum_{k=1}^{K} \left[ \left\langle g(z_{k-1/2}), z_{k-1/2} - z_k \right\rangle - \alpha V_{z_{k-1}}(z_k) \right]$$

for all $u \in \mathcal{Z}$. Note that since $z_0$ minimizes $r$, $V_{z_0}(u) = r(u) - r(z_0) \leq \Theta$ for all $u$. Therefore, maximizing the above display over $u$ and afterwards taking expectation gives

$$\mathbb{E} \max_{u \in \mathcal{Z}} \sum_{k=1}^{K} \left\langle g(z_{k-1/2}), z_{k-1/2} - u \right\rangle \leq \alpha \Theta + \sum_{k=1}^{K} \mathbb{E} \left[ \left\langle g(z_{k-1/2}), z_{k-1/2} - z_k \right\rangle - \alpha V_{z_{k-1}}(z_k) \right].$$

Finally, by Definition 1, $\mathbb{E} \left[ \left\langle g(z_{k-1/2}), z_{k-1/2} - z_k \right\rangle - \alpha V_{z_{k-1}}(z_k) \right] \leq \varepsilon$ for every $k$, and and the result follows by dividing by $K$ and using the bound (8). $\qquad\square$

### B.2  Proof of Proposition 2

**Proposition 2.** *Let $\alpha, L > 0$, let $w_0 \in \mathcal{Z}$ and let $\tilde{g}_{w_0}$ be $(w_0, L)$-centered for monotone $g$. Then, for $\eta = \frac{\alpha}{10L^2}$ and $T \geq \frac{4}{\eta\alpha} = \frac{40L^2}{\alpha^2}$, the iterates of Algorithm 2 satisfy*

$$\mathbb{E} \max_{u \in \mathcal{Z}} \left[ \frac{1}{T} \sum_{t \in [T]} \langle g(w_t), w_t - u \rangle - \alpha V_{w_0}(u) \right] \leq 0. \tag{11}$$

*Proof.* Recall the expression $w_t = \arg\min_{w \in \mathcal{Z}} \left\{ \langle \eta \tilde{g}_{w_0}(w_{t-1}), w \rangle + \frac{\eta\alpha}{2} V_{w_0}(w) + V_{w_{t-1}}(w) \right\}$ for the iterates of Algorithm 2. We apply Lemma 4 with $Q(z) = \langle g(w_0), z \rangle + \frac{\alpha}{2} V_{w_0}(z)$ and $\gamma_t = \eta \tilde{\delta}_t$, where

$$\tilde{\delta}_t = \tilde{g}_{w_0}(w_t) - g(w_0).$$

Dividing through by $\eta$, the resulting regret bound reads

$$\sum_{t\in[T]} \left\langle \tilde{g}_{w_0}(w_t) + \tfrac{\alpha}{2}\nabla V_{w_0}(w_t), w_t - u \right\rangle \leq \frac{V_{w_0}(u)}{\eta} + \frac{\eta}{2}\sum_{t\in[T]} \|\tilde{\delta}_t\|_*^2, \qquad (20)$$

where we used the fact that $\tilde{\delta}_0 = 0$ to drop the summation over $t = 0$ in the RHS. Now, let

$$\tilde{\Delta}_t = g(w_t) - \tilde{g}_{w_0}(w_t).$$

Rearranging the inequality (20), we may write it as

$$\sum_{t\in[T]} \left\langle g(w_t) + \tfrac{\alpha}{2}\nabla V_{w_0}(w_t), w_t - u \right\rangle \leq \frac{V_{w_0}(u)}{\eta} + \frac{\eta}{2}\sum_{t\in[T]} \|\tilde{\delta}_t\|_*^2 + \sum_{t\in[T]} \left\langle \tilde{\Delta}_t, w_t - u \right\rangle. \qquad (21)$$

Define the "ghost iterate" sequence $s_1, s_2, \ldots, s_T$ according to

$$s_t = \arg\min_{s\in\mathcal{Z}} \left\{ \left\langle \eta\tilde{\Delta}_{t-1}, s \right\rangle + V_{s_{t-1}}(s) \right\} \quad \text{with} \ \ s_0 = w_0.$$

Applying Lemma 4 with $Q = 0$ and $\gamma_t = \eta\tilde{\Delta}_t$, we have

$$\sum_{t\in[T]} \left\langle \tilde{\Delta}_t, s_t - u \right\rangle \leq \frac{V_{w_0}(u)}{\eta} + \frac{\eta}{2}\sum_{t\in[T]} \|\tilde{\Delta}_t\|_*^2, \qquad (22)$$

where here too we used $\tilde{\Delta}_0 = 0$. Writing $\left\langle \tilde{\Delta}_t, w_t - u \right\rangle = \left\langle \tilde{\Delta}_t, w_t - s_t \right\rangle + \left\langle \tilde{\Delta}_t, s_t - u \right\rangle$ and substituting (22) into (21) we have

$$\sum_{t\in[T]} \left\langle g(w_t) + \tfrac{\alpha}{2}\nabla V_{w_0}(w_t), w_t - u \right\rangle \leq \frac{2V_{w_0}(u)}{\eta} + \frac{\eta}{2}\sum_{t\in[T]} \left[\|\tilde{\delta}_t\|_*^2 + \|\tilde{\Delta}_t\|_*^2\right] + \sum_{t\in[T]} \left\langle \tilde{\Delta}_t, w_t - s_t \right\rangle.$$

Substituting

$$-\tfrac{\alpha}{2}\left\langle \nabla V_{w_0}(w_t), w_t - u \right\rangle = \tfrac{\alpha}{2}V_{w_0}(u) - \tfrac{\alpha}{2}V_{w_t}(u) - \tfrac{\alpha}{2}V_{w_0}(w_t) \leq \tfrac{\alpha}{2}V_{w_0}(u) - \tfrac{\alpha}{2}V_{w_0}(w_t)$$

and dividing by $T$, we have

$$\frac{1}{T}\sum_{t\in[T]} \left\langle g(w_t), w_t - u \right\rangle \leq \left(\tfrac{2}{\eta T} + \tfrac{\alpha}{2}\right)V_{w_0}(u) + \frac{1}{T}\sum_{t\in[T]} \left[\tfrac{\eta}{2}\|\tilde{\delta}_t\|_*^2 + \tfrac{\eta}{2}\|\tilde{\Delta}_t\|_*^2 - \tfrac{\alpha}{2}V_{w_0}(w_t) + \left\langle \tilde{\Delta}_t, w_t - s_t \right\rangle\right].$$

Subtracting $\alpha V_{w_0}(u)$ from both sides and using $\tfrac{2}{\eta T} - \tfrac{\alpha}{2} \leq 0$ due to $T \geq \tfrac{4}{\eta\alpha}$, we obtain

$$\frac{1}{T}\sum_{t\in[T]} \left\langle g(w_t), w_t - u \right\rangle - \alpha V_{w_0}(u) \leq \frac{1}{T}\sum_{t\in[T]} \left[\tfrac{\eta}{2}\|\tilde{\delta}_t\|_*^2 + \tfrac{\eta}{2}\|\tilde{\Delta}_t\|_*^2 - \tfrac{\alpha}{2}V_{w_0}(w_t) + \left\langle \tilde{\Delta}_t, w_t - s_t \right\rangle\right].$$

Note that this inequality holds with probability 1 for all $u$. We may therefore maximize over $u$ and then take expectation, obtaining

$$\mathbb{E}\max_{u\in\mathcal{Z}} \left\{ \frac{1}{T}\sum_{t\in[T]} \left\langle g(w_t), w_t - u \right\rangle - \alpha V_{w_0}(u) \right\}$$

$$\leq \frac{1}{T}\sum_{t\in[T]} \mathbb{E}\left[\tfrac{\eta}{2}\|\tilde{\delta}_t\|_*^2 + \tfrac{\eta}{2}\|\tilde{\Delta}_t\|_*^2 - \tfrac{\alpha}{2}V_{w_0}(w_t) + \left\langle \tilde{\Delta}_t, w_t - s_t \right\rangle\right]. \qquad (23)$$

It remains to argue the the RHS is nonpositive. By the first centered estimator property, we have

$$\mathbb{E}\left[\tilde{\Delta}_t \mid w_t, s_t\right] = \mathbb{E}\left[g(w_t) - \tilde{g}_{w_0}(w_t) \mid w_t, s_t\right] = 0$$

and therefore $\mathbb{E}\left\langle \tilde{\Delta}_t, w_t - s_t \right\rangle = 0$ for all $t$. By the second property

$$\mathbb{E}\|\tilde{\delta}_t\|_*^2 = \mathbb{E}\|\tilde{g}_{w_0}(w_t) - g(w_0)\|_*^2 \leq L^2\|w_t - w_0\|^2 \leq 2L^2 V_{w_0}(w_t),$$

where the last transition used the strong convexity of $r$. Similarly, by Lemma 1 we have

$$\mathbb{E}\|\tilde{\Delta}_t\|_*^2 = \mathbb{E}\|\tilde{g}_{w_0}(w_t) - g(w)\|_*^2 \leq 4L^2\|w_t - w_0\|^2 \leq 8L^2 V_{w_0}(w_t).$$

Therefore

$$\mathbb{E}\left[\tfrac{\eta}{2}\|\tilde{\delta}_t\|_*^2 + \tfrac{\eta}{2}\|\tilde{\Delta}_t\|_*^2 - \tfrac{\alpha}{2}V_{w_0}(w_t)\right] \leq (5\eta L^2 - \tfrac{\alpha}{2})\mathbb{E}V_{w_0}(w_t) = 0,$$

using $\eta = \frac{\alpha}{10L^2}$. $\qquad\qquad\square$

# C The $\ell_1$-$\ell_1$ setup

## C.1 Complete pseudo-code

---

**Algorithm 3:** Variance reduction for $\ell_1$-$\ell_1$ games

---

**Input**: Matrix $A \in \mathbb{R}^{m \times n}$ with $i$th row $A_{i:}$ and $j$th column $A_{:j}$, target accuracy $\epsilon$
**Output**: A point with expected duality gap below $\epsilon$

1   $L \leftarrow \max_{ij} |A_{ij}|, \alpha \leftarrow L\sqrt{\frac{n+m}{\text{nnz}(A)}}, K \leftarrow \left\lceil \frac{\log(nm)\alpha}{\epsilon} \right\rceil, \eta \leftarrow \frac{\alpha}{10L^2}, T \leftarrow \left\lceil \frac{4}{\eta\alpha} \right\rceil, z_0 \leftarrow (\frac{1}{n}\mathbf{1}_n, \frac{1}{m}\mathbf{1}_m)$

2   **for** $k = 1, \dots, K$ **do**

     ▷ *Relaxed oracle query:*

3      $(x_0, y_0) \leftarrow (z_{k-1}^{\mathsf{x}}, z_{k-1}^{\mathsf{y}}), (g_0^{\mathsf{x}}, g_0^{\mathsf{y}}) \leftarrow (A^\top y_0, -Ax_0)$

4      **for** $t = 1, \dots, T$ **do**

         ▷ *Gradient estimation:*

5          Sample $i \sim p$ where $p_i = \frac{|[y_{t-1}]_i - [y_0]_i|}{\|y_{t-1} - y_0\|_1}$, sample $j \sim q$ where $q_j = \frac{|[x_{t-1}]_j - [x_0]_j|}{\|x_{t-1} - x_0\|_1}$

6          Set $\tilde{g}_{t-1} = g_0 + \left( A_{i:} \frac{[y_{t-1}]_i - [y_0]_i}{p_i}, -A_{:j} \frac{[x_{t-1}]_j - [x_0]_j}{q_j} \right)$

         ▷ *Mirror descent step:*

7          $x_t \leftarrow \Pi_{\mathcal{X}} \left( \frac{1}{1 + \eta\alpha/2} \left( \log x_{t-1} + \frac{\eta\alpha}{2} \log x_0 - \eta\tilde{g}_{t-1}^{\mathsf{x}} \right) \right)$      ▷ $\Pi_{\mathcal{X}}(v) = \frac{e^v}{\|e^v\|_1}$

8          $y_t \leftarrow \Pi_{\mathcal{Y}} \left( \frac{1}{1 + \eta\alpha/2} \left( \log y_{t-1} + \frac{\eta\alpha}{2} \log y_0 - \eta\tilde{g}_{t-1}^{\mathsf{y}} \right) \right)$      ▷ $\Pi_{\mathcal{Y}}(v) = \frac{e^v}{\|e^v\|_1}$

9      $z_{k-1/2} \leftarrow \frac{1}{T} \sum_{t=1}^{T} (x_t, y_t)$

     ▷ *Extragradient step:*

10      $z_k^{\mathsf{x}} \leftarrow \Pi_{\mathcal{X}} \left( \log z_{k-1}^{\mathsf{x}} - \frac{1}{\alpha} A^\top z_{k-1/2}^{\mathsf{y}} \right)$

11      $z_k^{\mathsf{y}} \leftarrow \Pi_{\mathcal{Y}} \left( \log z_{k-1}^{\mathsf{y}} + \frac{1}{\alpha} A z_{k-1/2}^{\mathsf{x}} \right)$

12   **return** $\frac{1}{K} \sum_{k=1}^{K} z_{k-1/2}$

---

## C.2 Proof of runtime bound

**Theorem 1.** *Let $A \in \mathbb{R}^{m \times n}$, $\epsilon > 0$, and $\alpha \geq \epsilon / \log(nm)$. Algorithm 3 outputs a point $z = (z^{\mathsf{x}}, z^{\mathsf{y}})$ such that $\mathbb{E}\left[ \max_{y \in \Delta^m} y^\top A z^{\mathsf{x}} - \min_{x \in \Delta^n} (z^{\mathsf{y}})^\top A x \right] = \mathbb{E}\left[ \max_i [Az^{\mathsf{x}}]_i - \min_j [A^\top z^{\mathsf{y}}]_j \right] \leq \epsilon$, and runs in time*

$$O\left( \left( \text{nnz}(A) + \frac{(m+n)\|A\|_{\max}^2}{\alpha^2} \right) \frac{\alpha \log(mn)}{\epsilon} \right). \tag{15}$$

*Setting $\alpha$ optimally, the running time is*

$$O\left( \text{nnz}(A) + \frac{\sqrt{\text{nnz}(A)(m+n)}\|A\|_{\max}\log(mn)}{\epsilon} \right). \tag{16}$$

*Proof.* First, we prove the expected duality gap bound. By Lemma 2 and Corollary 1 (with $L = \|A\|_{\max}$), InnerLoop is an $(\alpha, 0)$-relaxed proximal oracle. On $\Delta^d$, negative entropy has minimum value $-\log d$ and is non-positive, therefore for the $\ell_1$-$\ell_1$ domain we have $\Theta = \max_{z'} r(z') - \min_z r(z) = \log(nm)$. By Proposition 1, running $K \geq \alpha \log(nm)/\epsilon$ iterations guarantees an $\epsilon$-approximate saddle point in expectation.

Now, we prove the runtime bound. Lines 3, 10 and 11 of Algorithm 3 each take time $O(\text{nnz}(A))$, as they involve matrix-vector products with $A$ and $A^\top$. All other lines run in time $O(n + m)$, as they consist of sampling and vector arithmetic (the time to compute sampling probabilities dominates the runtime of sampling). Therefore, the total runtime is $O((\text{nnz}(A) + (n + m)T)K)$. Substituting $T \leq 1 + \frac{40L^2}{\alpha^2}$ and $K \leq 1 + \frac{\log(nm)\alpha}{\epsilon}$ gives the bound (15). Setting

$$\alpha = \max\left\{\frac{\epsilon}{\log nm}\,,\, \|A\|_{\max}\sqrt{\frac{n + m}{\text{nnz}(A)}}\right\}$$

gives the optimized bound (16). $\qquad\square$

**Remark 3.** We can improve the $\log(mn)$ factor in (15) and (16) to $\sqrt{\log m \log n}$ by the transformation $\mathcal{X} \to \mathcal{X}\sqrt{\frac{\log m}{\log n}}$ and $\mathcal{Y} \to \mathcal{Y}\sqrt{\frac{\log n}{\log m}}$. This transformation leaves the problem unchanged and reduces $\Theta$ from $\log(mn)$ to $2\sqrt{\log m \log n}$. It is also equivalent to proportionally using slightly different step-sizes for the $\mathcal{X}$ and $\mathcal{Y}$ block.

# D  The $\ell_2$-$\ell_1$ setup

## D.1  Derivation of gradient clipping

### D.1.1  Basic gradient estimator

We first present a straightforward adaptation of the $\ell_1$-$\ell_1$ gradient estimator, which we subsequently improve to obtain the optimal Lipschitz constant dependence. Following the "sampling from the difference" strategy, consider a gradient estimator $\tilde{g}_{w_0}$ computed as in (13), but with the following different choice of $q(w)$:

$$p_i(w) = \frac{|[w^{\mathsf{y}}]_i - [w_0^{\mathsf{y}}]_i|}{\|w^{\mathsf{y}} - w_0^{\mathsf{y}}\|_1} \quad \text{and} \quad q_j(w) = \frac{([w^{\mathsf{x}}]_j - [w_0^{\mathsf{x}}]_j)^2}{\|w^{\mathsf{x}} - w_0^{\mathsf{x}}\|_2^2}. \tag{24}$$

The resulting gradient estimator has the explicit form

$$\tilde{g}_{w_0}(w) = g(w_0) + \left(A_{i:}\frac{\|w^{\mathsf{y}} - w_0^{\mathsf{y}}\|_1}{\text{sign}([w^{\mathsf{y}} - w_0^{\mathsf{y}}]_i)}, -A_{:j}\frac{\|w^{\mathsf{x}} - w_0^{\mathsf{x}}\|_2^2}{[w^{\mathsf{x}} - w_0^{\mathsf{x}}]_j}\right). \tag{25}$$

(Note that $\tilde{g}_{w_0}$ of the form (13) is finite with probability 1.) Direct calculation shows it is centered.

**Lemma 5.** *In the $\ell_2$-$\ell_1$ setup, the estimator (25) is $(w_0, L)$-centered with $L = \sqrt{\sum_{j\in[n]}\|A_{:j}\|_\infty^2}$.*

*Proof.* The estimator is unbiased since it is of the form (13). To show the variance bound, first consider the $\mathcal{X}$-block. We have

$$\left\|\tilde{g}_{w_0}^{\mathsf{x}}(w) - g^{\mathsf{x}}(w_0)\right\|_2^2 = \|A_{i:}\|_2^2\|w^{\mathsf{y}} - w_0^{\mathsf{y}}\|_1^2 \leq \|A\|_{2\to\infty}^2\|w^{\mathsf{y}} - w_0^{\mathsf{y}}\|_1^2 \leq L^2\|w^{\mathsf{y}} - w_0^{\mathsf{y}}\|_1^2, \tag{26}$$

where we used $\|A\|_{2\to\infty}^2 = \max_{i\in[n]}\|A_{i:}\|_2^2 \leq \sum_{j\in[m]}\|A_{:j}\|_\infty^2 = L^2$. Second, for the $\mathcal{Y}$-block,

$$\mathbb{E}\left\|\tilde{g}_{w_0}^{\mathsf{y}}(w) - g^{\mathsf{y}}(w_0)\right\|_\infty^2 = \sum_{j\in[n]}\frac{\|A_{:j}\|_\infty^2[w^{\mathsf{x}} - w_0^{\mathsf{x}}]_j^2}{q_j(w)} = L^2\|w^{\mathsf{x}} - w_0^{\mathsf{x}}\|_2^2. \tag{27}$$

Combining (26) and (27), we have the second property $\mathbb{E}\|\tilde{g}_{w_0}(w) - g(w_0)\|_*^2 \leq L^2\|w - w_0\|^2$. $\quad\square$

### D.1.2  Improved gradient estimator

The constant $L$ in Lemma 5 is larger than the Lipschitz constant of $g$ (i.e. $\|A\|_{2\to\infty}$) by a factor of up to $\sqrt{n}$. Consequently, a variance reduction scheme based on the estimator (25) will not always improve on the linear-time mirror prox method.

Inspecting the proof of Lemma 5, we see that the cause for the inflated value of $L$ is the bound (27) on $\mathbb{E} \left\| \tilde{g}^{\mathsf{y}}_{w_0}(w) - g^{\mathsf{y}}(w_0) \right\|^2_\infty$. We observe that swapping the order of expectation and maximization would solve the problem, as

$$\max_{k \in [m]} \mathbb{E} \left[ \tilde{g}^{\mathsf{y}}_{w_0}(w) - g^{\mathsf{y}}(w_0) \right]^2_k = \max_{k \in [m]} \sum_{j \in [n]} \frac{A^2_{kj} [w^{\mathsf{x}} - w^{\mathsf{x}}_0]^2_j}{q_j(w)} = \|A\|^2_{2 \to \infty} \|w^{\mathsf{x}} - w^{\mathsf{x}}_0\|^2_2. \qquad (28)$$

Moreover, inspecting the proof of Proposition 2 reveals that instead of bounding terms of the form $\mathbb{E} \left\| \tilde{g}^{\mathsf{y}}_{w_0}(w_t) - g^{\mathsf{y}}(w_0) \right\|^2_\infty$ we may directly bound $\mathbb{E} \left[ \eta \langle \tilde{g}^{\mathsf{y}}_{w_0}(w_t) - g^{\mathsf{y}}(w_0), y_t - y_{t+1} \rangle - V_{y_t}(y_{t+1}) \right]$, where we write $w_t = (x_t, y_t)$ and recall that $\eta$ is the step-size in Algorithm 2. Suppose that $\eta \left\| \tilde{g}^{\mathsf{y}}_{w_0}(w_t) - g^{\mathsf{y}}(w_0) \right\|_\infty \leq 1$ holds. In this case we may use a "local norms" bound (Lemma **??** in Appendix D.2) to write

$$\eta \langle \tilde{g}^{\mathsf{y}}_{w_0}(w_t) - g^{\mathsf{y}}(w_0), y_t - y_{t+1} \rangle - V_{y_t}(y_{t+1}) \leq \eta^2 \sum_{k \in [m]} [y_t]_k [\tilde{g}^{\mathsf{y}}_{w_0}(w_t) - g^{\mathsf{y}}(w_0)]^2_k$$

and bound the expectation of the RHS using (28) conditional on $w_t$.

Unfortunately, the gradient estimator (25) does not always satisfy $\eta \left\| \tilde{g}^{\mathsf{y}}_{w_0}(w_t) - g^{\mathsf{y}}(w_0) \right\|_\infty \leq 1$. Following Clarkson et al. [6], we enforce this bound by clipping the gradient estimates, yielding the estimator

$$\tilde{g}_{w_0}(w) := \left( A^\top w^{\mathsf{y}}_0 + A_{i:} \frac{[w^{\mathsf{y}}]_i - [w^{\mathsf{y}}_0]_i}{p_i(w)}, -A w^{\mathsf{x}}_0 - \mathsf{T}_\tau \left( A_{:j} \frac{[w^{\mathsf{x}}]_j - [w^{\mathsf{x}}_0]_j}{q_j(w)} \right) \right),$$

$$\text{where } [\mathsf{T}_\tau(v)]_i = \begin{cases} -\tau & [v]_i < -\tau \\ [v]_i & -\tau \leq [v]_i \leq \tau \\ \tau & [v]_i > \tau, \end{cases} \qquad (29)$$

where $i \sim p(w)$ and $j \sim q(w)$ with $p, q$ as defined in (24). The clipping in (29) does not significantly change the variance of the estimator, but it introduces some bias for which we must account. We summarize the relevant properties of the clipped gradient estimator in the following.

**Definition 3.** *Let $w_0 = (w^{\mathsf{x}}_0, w^{\mathsf{y}}_0) \in \mathcal{Z}$ and $\tau, L > 0$. A stochastic gradient estimator $\tilde{g}_{w_0} : \mathcal{Z} \to \mathcal{Z}^*$ is called $(w_0, L, \tau)$-centered-bounded-biased (CBB) if it satisfies for all $w = (w^{\mathsf{x}}, w^{\mathsf{y}}) \in \mathcal{Z}$,*

1. *$\mathbb{E}\tilde{g}^{\mathsf{x}}_{w_0}(w) = g^{\mathsf{x}}(w)$ and $\left\| \mathbb{E}\tilde{g}^{\mathsf{y}}_{w_0}(w) - g^{\mathsf{y}}(w) \right\|_* \leq \frac{L^2}{\tau} \|w - w_0\|^2$,*

2. *$\left\| \tilde{g}^{\mathsf{y}}_{w_0}(w) - g^{\mathsf{y}}(w_0) \right\|_* \leq \tau$ and $\left\| \tilde{g}^{\mathsf{y}}_{w_0}(w) - g^{\mathsf{y}}(w) \right\|_* \leq 2L + \tau$,*

3. *$\mathbb{E} \left\| \tilde{g}^{\mathsf{x}}_{w_0}(w) - g^{\mathsf{x}}(w_0) \right\|^2_* + \max_{i \in [m]} \mathbb{E} \left[ \tilde{g}^{\mathsf{y}}_{w_0}(w) - g^{\mathsf{y}}(w_0) \right]^2_i \leq L^2 \|w - w_0\|^2.$*

**Lemma 6.** *In the $\ell_2$-$\ell_1$ setup, the estimator (29) is $(w_0, L, \tau)$-CBB with $L = \|A\|_{2 \to \infty}$.*

*Proof.* The $\mathcal{X}$ component for the gradient estimator is unbiased. We bound the bias in the $\mathcal{Y}$ block as follows. Fixing an index $i \in [m]$, we have

$$\left| \mathbb{E} \left[ \tilde{g}^{\mathsf{y}}_{w_0}(w) - g^{\mathsf{y}}(w) \right]_i \right| = \left| \mathbb{E}_j \left[ A_{ij} \frac{[w^{\mathsf{x}}]_j - [w^{\mathsf{x}}_0]_j}{q_j} - \mathsf{T}_\tau \left( A_{ij} \frac{[w^{\mathsf{x}}]_j - [w^{\mathsf{x}}_0]_j}{q_j} \right) \right] \right|$$

$$\leq \sum_{j \in \mathcal{J}_\tau(i)} q_j \left| A_{ij} \frac{[w^{\mathsf{x}}]_j - [w^{\mathsf{x}}_0]_j}{q_j} - \mathsf{T}_\tau \left( A_{ij} \frac{[w^{\mathsf{x}}]_j - [w^{\mathsf{x}}_0]_j}{q_j} \right) \right|$$

$$\leq \sum_{j \in \mathcal{J}_\tau(i)} |A_{ij}| \, |[w^{\mathsf{x}}]_j - [w^{\mathsf{x}}_0]_j|$$

where the last transition used $|a - \mathsf{T}_\tau(a)| \leq |a|$ for all $a$, and

$$\mathcal{J}_\tau(i) = \left\{ j \in [n] \mid \mathsf{T}_\tau \left( A_{ij} \frac{[w^{\mathsf{x}}]_j - [w^{\mathsf{x}}_0]_j}{q_j} \right) \neq A_{ij} \frac{[w^{\mathsf{x}}]_j - [w^{\mathsf{x}}_0]_j}{q_j} \right\}.$$

Note that $j \in \mathcal{J}_\tau(i)$ if and only if

$$\left| A_{ij} \frac{[w^\times]_j - [w_0^\times]_j}{q_j} \right| = \frac{\|w^\times - w_0^\times\|_2^2 |A_{ij}|}{|[w^\times]_j - [w_0^\times]_j|} > \tau \Rightarrow |[w^\times]_j - [w_0^\times]_j| \leq \frac{1}{\tau} \|w^\times - w_0^\times\|_2^2 |A_{ij}|.$$

Therefore,

$$\sum_{j \in \mathcal{J}_\tau(i)} |A_{ij}| |[w^\times]_j - [w_0^\times]_j| \leq \frac{1}{\tau} \|w^\times - w_0^\times\|_2^2 \sum_{j \in \mathcal{J}_\tau(i)} |A_{ij}|^2 \leq \frac{1}{\tau} \|w^\times - w_0^\times\|_2^2 \|A_{i:}\|_2^2$$

and $\left\|\mathbb{E}\tilde{g}_{w_0}^{\mathsf{y}}(w) - g^{\mathsf{y}}(w)\right\|_\infty \leq \frac{L^2}{\tau} \|w^\times - w_0^\times\|_2^2$ follows by taking the maximum over $i \in [m]$.

By definition of $\mathsf{T}_\tau$ we have $\left\|\tilde{g}_{w_0}^{\mathsf{y}}(w) - g^{\mathsf{y}}(w_0)\right\|_\infty \leq \tau$ and by the triangle inequality and $L$-Lipschitz continuity of $g$ we have

$$\left\|\tilde{g}_{w_0}^{\mathsf{y}}(w) - g^{\mathsf{y}}(w)\right\|_\infty \leq \|g^{\mathsf{y}}(w) - g^{\mathsf{y}}(w_0)\|_\infty + \left\|\tilde{g}_{w_0}^{\mathsf{y}}(w) - g^{\mathsf{y}}(w_0)\right\|_\infty \leq L \|w^\times - w_0^\times\|_2 + \tau \leq 2L + \tau,$$

since we assume $\mathcal{X}$ is the unit Euclidean ball.

Finally, we note that for all $k$, the addition of $\mathsf{T}_\tau$ never increases $[\tilde{g}_{w_0}^{\mathsf{y}}(w) - g^{\mathsf{y}}(w_0)]_k^2$, and so the third property follows from (28) and (26). $\qquad\square$

To guarantee $\eta \left\|\tilde{g}_{w_0}^{\mathsf{y}}(w_t) - g^{\mathsf{y}}(w_0)\right\|_\infty \leq 1$, we set the threshold $\tau$ to be $1/\eta$. By the first property in Definition 3, the bias caused by this choice of $\tau$ is of the order of the variance of the estimator, and we may therefore cancel it with the regularizer by choosing $\eta$ slightly smaller than in Proposition 2. In Appendix D we prove (using the observations from the preceding discussion) that Algorithm 2 with a CBB gradient estimator implements a relaxed proximal oracle.

**Proposition 3.** *In the $\ell_2$-$\ell_1$ setup, let $\alpha, L > 0$, let $w_0 \in \mathcal{Z}$ and let $\tilde{g}_{w_0}$ be $(w_0, L, \frac{20L^2}{\alpha})$-CBB for monotone $g$. Then, for $\alpha \leq L$, $\eta = \frac{\alpha}{20L^2}$ and $T \geq \frac{4}{\eta\alpha} = \frac{80L^2}{\alpha^2}$, the iterates of Algorithm 2 satisfy the bound* (11). *Moreover, for $g(z) = (A^\top z^{\mathsf{y}}, -Az^\times)$, $\mathcal{O}(w_0) = \text{InnerLoop}(w_0, \tilde{g}_{w_0}, \alpha)$ is an $(\alpha, 0)$-relaxed proximal oracle.*

We remark that the proof of Proposition 3 relies on the structure of the simplex with negative entropy as the distance generating function. For this reason, we state the proposition for the $\ell_2$-$\ell_1$ setup. However, Proposition 3 would also hold for other setups where $\mathcal{Y}$ is the simplex and $r^{\mathsf{y}}$ is the negative entropy, provided a CBB gradient estimator is available.

With Proposition 3 in hand, the proof of Theorem 2 follows identically to that of Theorem 1, except Proposition 3 replaces Corollary 1, $L$ is now $\|A\|_{2\to\infty}$ instead of $\|A\|_{\max}$, and $\Theta = \max_{z'} r(z') - \min_z r(z) = \frac{1}{2} + \log m \leq \log(2m)$ rather than $\log(mn)$.

Before giving the proof of Proposition 3 is Section D.4, we first collect some properties of the KL divergence (Section D.2) and of centered-bounded-biased (CBB) gradient estimators (Section D.3).

### D.2 Local norms bounds

For this subsection, let $\mathcal{Y}$ be the $m$ dimensional simplex $\Delta^m$, and let $r(y) = \sum_{i=1}^m y_i \log y_i$ be the negative entropy distance generating function. The corresponding Bregman divergence is the KL divergence, which is well-defined for any $y, y' \in \mathbb{R}_{\geq 0}^m$ and has the form

$$V_y(y') = \sum_{i \in [m]} \left[ y_i' \log \frac{y_i'}{y_i} + y_i - y_i' \right] = \int_0^1 dt \int_0^t \sum_{i \in [m]} \frac{(y_i - y_i')^2}{(1 - \tau)y_i + \tau y_i'} d\tau. \tag{30}$$

In the literature, "local norms" regret analysis [37, Section 2.8] relies on the fact that $r^*(\gamma) = \log(\sum_i e^{\gamma_i})$ (the conjugate of negative entropy in the simplex) is locally smooth with respect to a Euclidean norm weighted by $\nabla r^*(\gamma) = \frac{e^\gamma}{\|e^\gamma\|_1}$. More precisely, the Bregman divergence $V_\gamma^*(\gamma') = r^*(\gamma') - r^*(\gamma) - \langle \nabla r^*(\gamma), \gamma' - \gamma \rangle$ satisfies

$$V_\gamma^*(\gamma + \delta) \leq \|\delta\|_{\nabla r^*(\gamma)}^2 := \sum_i [\nabla r^*(\gamma)]_i \cdot \delta_i^2 \quad \text{whenever } \delta_i \leq 1.79 \ \forall i. \tag{31}$$

Below, we state this bound in a form that is directly applicable to our analysis.

**Lemma 7.** *Let $y, y' \in \Delta^m$ and $\delta \in \mathbb{R}^m$. If $\delta$ satisfies $\delta_i \leq 1.79$ for all $i \in [m]$ then the KL divergence $V_y(y')$ satisfies*

$$\langle \delta, y' - y \rangle - V_y(y') \leq \|\delta\|_y^2 := \sum_{i \in [m]} y_i \delta_i^2$$

*Proof.* It suffices to consider $y$ in the relative interior of the simplex where $r$ is differentiable; the final result will hold for any $y$ in the simplex by continuity. Recall the following general facts about convex conjugates: $\langle \gamma', y' \rangle - r(y') \leq r^*(\gamma')$ for any $\gamma' \in \mathbb{R}^m$, $y = \nabla r^*(\nabla r(y))$ and $r^*(\nabla r(y)) = \langle \nabla r(y), y \rangle - r(y)$. Therefore, we have for all $y' \in \Delta^m$,

$$\langle \delta, y' - y \rangle - V_y(y') = \langle \nabla r(y) + \delta, y' \rangle - r(y') - [\langle \nabla r(y), y \rangle - r(y)] - \langle y, \delta \rangle$$
$$\leq r^*(\nabla r(y) + \delta) - r^*(\nabla r(y)) - \langle \nabla r^*(\nabla r(y)), \delta \rangle = V_{\nabla r(y)}^*(\nabla r(y) + \delta).$$

The result follows from (31) with $\gamma = \nabla r(y)$, recalling again that $y = \nabla r^*(\nabla r(y))$. For completeness we prove (31) below, following [37]. We have

$$r^*(\gamma + \delta) - r^*(\gamma) = \log \left( \frac{\sum_{i \in [m]} e^{\gamma_i + \delta_i}}{\sum_{i \in [m]} e^{\gamma_i}} \right) \overset{(i)}{\leq} \log \left( 1 + \frac{\sum_{i \in [m]} e^{\gamma_i}(\delta_i + \delta_i^2)}{\sum_{i \in [m]} e^{\gamma_i}} \right)$$

$$= \log(1 + \langle \nabla r^*(\gamma), \delta + \delta^2 \rangle) \overset{(ii)}{\leq} \langle \nabla r^*(\gamma), \delta \rangle + \langle \nabla r^*(\gamma), \delta^2 \rangle,$$

where $(i)$ follows from $e^x \leq 1 + x + x^2$ for all $x \leq 1.79$ and $(ii)$ follows from $\log(1 + x) \leq x$ for all $x$. Therefore,

$$V_\gamma^*(\gamma + \delta) = r^*(\gamma + \delta) - r^*(\gamma) - \langle \nabla r^*(\gamma), \delta \rangle \leq \langle \nabla r^*(\gamma), \delta^2 \rangle = \|\delta\|_{\nabla r^*(\gamma)}^2,$$

completing the proof. $\qquad\square$

### D.3 Properties of CBB gradient estimators

We recall the definition of a centered-bounded-biased gradient estimator.

**Definition 3.** *Let $w_0 = (w_0^{\mathsf{x}}, w_0^{\mathsf{y}}) \in \mathcal{Z}$ and $\tau, L > 0$. A stochastic gradient estimator $\tilde{g}_{w_0} : \mathcal{Z} \to \mathcal{Z}^*$ is called $(w_0, L, \tau)$-centered-bounded-biased (CBB) if it satisfies for all $w = (w^{\mathsf{x}}, w^{\mathsf{y}}) \in \mathcal{Z}$,*

1. *$\mathbb{E}\tilde{g}_{w_0}^{\mathsf{x}}(w) = g^{\mathsf{x}}(w)$ and $\left\| \mathbb{E}\tilde{g}_{w_0}^{\mathsf{y}}(w) - g^{\mathsf{y}}(w) \right\|_* \leq \frac{L^2}{\tau} \|w - w_0\|^2$,*

2. *$\left\| \tilde{g}_{w_0}^{\mathsf{y}}(w) - g^{\mathsf{y}}(w_0) \right\|_* \leq \tau$ and $\left\| \tilde{g}_{w_0}^{\mathsf{y}}(w) - g^{\mathsf{y}}(w) \right\|_* \leq 2L + \tau$,*

3. *$\mathbb{E} \left\| \tilde{g}_{w_0}^{\mathsf{x}}(w) - g^{\mathsf{x}}(w_0) \right\|_*^2 + \max_{i \in [m]} \mathbb{E} [\tilde{g}_{w_0}^{\mathsf{y}}(w) - g^{\mathsf{y}}(w_0)]_i^2 \leq L^2 \|w - w_0\|^2$.*

CBB estimators have the following additional property, analogous to Lemma 1.

**Lemma 8.** *In the $\ell_2$-$\ell_1$ setup, a $(w_0, L, \tau)$-CBB estimator with for $g$ with $\tau \geq 2\sqrt{2}L$ also satisfies, for all $w \in \mathcal{Z}$,*

$$\mathbb{E} \left\| \tilde{g}_{w_0}^{\mathsf{x}}(w) - g^{\mathsf{x}}(w) \right\|_2^2 + \max_{i \in [m]} \mathbb{E} [\tilde{g}_{w_0}^{\mathsf{y}}(w) - g^{\mathsf{y}}(w)]_i^2 \leq 2L^2 \|w - w_0\|^2.$$

*Proof.* We have $\mathbb{E} \left\| \tilde{g}_{w_0}^{\mathsf{x}}(w) - g^{\mathsf{x}}(w) \right\|_2^2 \leq \mathbb{E} \left\| \tilde{g}_{w_0}^{\mathsf{x}}(w) - g^{\mathsf{x}}(w_0) \right\|_2^2$ since the $\mathcal{X}$ component is unbiased. For the $\mathcal{Y}$ component, fix $i \in [m]$ and write

$$\mathbb{E} [\tilde{g}_{w_0}^{\mathsf{y}}(w) - g^{\mathsf{y}}(w)]_i^2 = \mathbb{E} [\tilde{g}_{w_0}^{\mathsf{y}}(w) - \mathbb{E}\tilde{g}_{w_0}^{\mathsf{y}}(w)]_i^2 + [\mathbb{E}\tilde{g}_{w_0}^{\mathsf{y}}(w) - g^{\mathsf{y}}(w)]_i^2$$

$$\leq \mathbb{E} [\tilde{g}_{w_0}^{\mathsf{y}}(w) - g(w_0)]_i^2 + \left( \frac{L^2}{\tau} \|w - w_0\|^2 \right)^2,$$

where the last inequality follows from the first CBB property and the fact that $[v]_i^2 \leq \|v\|_\infty^2$. Using $\tau \geq 2\sqrt{2}L$ and $\|w - w_0\| \leq 2\sqrt{2}$ for every $w, w_0 \in \mathbb{B}^n \times \Delta^m$, we obtain the result. $\qquad\square$

### D.4 Proof of Proposition 3

**Proposition 3.** *In the $\ell_2$-$\ell_1$ setup, let $\alpha, L > 0$, let $w_0 \in \mathcal{Z}$ and let $\tilde{g}_{w_0}$ be $(w_0, L, \frac{20L^2}{\alpha})$-CBB for monotone $g$. Then, for $\alpha \leq L$, $\eta = \frac{\alpha}{20L^2}$ and $T \geq \frac{4}{\eta\alpha} = \frac{80L^2}{\alpha^2}$, the iterates of Algorithm 2 satisfy the bound* (11). *Moreover, for $g(z) = (A^\top z^y, -Az^x)$, $\mathcal{O}(w_0) = \mathtt{InnerLoop}(w_0, \tilde{g}_{w_0}, \alpha)$ is an $(\alpha, 0)$-relaxed proximal oracle.*

*Proof.* Let $w_1, ..., w_T$ denote the iterates of Algorithm 2 and let $w_{T+1} \equiv u$. We recall the following notation from the proof of Proposition 2: $\tilde{\delta}_t = \tilde{g}_{w_0}(w_t) - g(w_0)$, $\tilde{\Delta}_t = g(w_t) - \tilde{g}_{w_0}(w_t)$ and $s_t = \arg\min_{s \in \mathcal{Z}} \left\{ \langle \eta\tilde{\Delta}_{t-1}, s \rangle + V_{s_{t-1}}(s) \right\}$. Retracing the steps of the proof of Proposition 2 leading up to the bound (23), we observe that by using the first inequality in Lemma 4 rather than the second, the bound (23) becomes

$$\mathbb{E} \max_{u \in \mathcal{Z}} \left\{ \frac{1}{T} \sum_{t \in [T]} \langle g(w_t), w_t - u \rangle - \alpha V_{w_0}(u) \right\} \leq \frac{1}{T} \sum_{t \in [T]} \mathbb{E} \left[ -\frac{\alpha}{2} V_{w_0}(w_t) + \langle \tilde{\Delta}_t, w_t - s_t \rangle \right]$$

$$+ \frac{1}{\eta T} \sum_{t \in [T]} \mathbb{E} \left[ \langle \eta\tilde{\delta}_t, w_t - w_{t+1} \rangle - V_{w_t}(w_{t+1}) + \langle \eta\tilde{\Delta}_t, s_t - s_{t+1} \rangle - V_{s_t}(s_{t+1}) \right]. \quad (32)$$

Let us bound the various expectations in the RHS of (32) one by one. By the first CBB property, $E[\tilde{\Delta}_t^x \mid w_t, s_t] = 0$ and also $\left\| E[\tilde{\Delta}_t^y \mid w_t, s_t] \right\|_* \leq \frac{L^2}{\tau} \|w_t - w_0\|^2$. Consequently,

$$\mathbb{E} \langle \tilde{\Delta}_t, w_t - s_t \rangle \leq \frac{L^2}{\tau} \mathbb{E} \|w_t - w_0\|^2 \|w_t^y - s_t^y\|_1.$$

Using $\|y - y'\|_1 \leq 2$ for every $y, y' \in \mathcal{Y} = \Delta^m$ as well as $\tau = \frac{1}{\eta}$, we obtain

$$\mathbb{E} \langle \tilde{\Delta}_t, w_t - s_t \rangle \leq 2\eta L^2 \mathbb{E} \|w_t - w_0\|^2 \leq 4\eta L^2 \mathbb{E} V_{w_0}(w_t). \quad (33)$$

To bound the expectation of $\langle \eta\tilde{\delta}_t, w_t - w_{t+1} \rangle - V_{w_t}(w_{t+1})$, we write $w_t = (x_t, y_t)$, and note that for the $\ell_2$-$\ell_1$ setup the Bregman divergence is separable, i.e. $V_{w_t}(w_{t+1}) = V_{x_t}(x_{t+1}) + V_{y_t}(y_{t+1})$. For the $\mathcal{X}$ component, we proceed as in Lemma 4, and write

$$\langle \eta\tilde{\delta}_t^x, x_t - x_{t+1} \rangle - V_{x_t}(x_{t+1}) \leq \frac{\eta^2}{2} \|\tilde{\delta}_t^x\|_2^2.$$

For the $\mathcal{Y}$ component, we observe that

$$\|\eta\tilde{\delta}_t^y\|_\infty = \eta \|\tilde{g}_{w_0}^y(w_t) - g^y(w_0)\|_\infty \leq \eta\tau = 1$$

by the second CBB property and $\tau = \frac{1}{\eta}$. Therefore, we may apply Lemma 7 with $\delta = -\eta\tilde{\delta}_t^y$ and obtain

$$\langle \eta\tilde{\delta}_t^y, y_t - y_{t+1} \rangle - V_{y_t}(y_{t+1}) \leq \eta^2 \sum_{i \in [m]} [y_t]_i [\tilde{\delta}_t^y]_i^2.$$

Taking expectation and using the fact that $y_t$ is in the simplex gives

$$\mathbb{E} \left[ \langle \eta\tilde{\delta}_t^y, y_t - y_{t+1} \rangle - V_{y_t}(y_{t+1}) \right] \leq \eta^2 \mathbb{E} \max_{i \in [m]} \mathbb{E} \left[ [\tilde{\delta}_t^y]_i^2 \mid w_t \right].$$

The third CBB property reads $\mathbb{E} \left[ \|\tilde{\delta}_t^x\|_2^2 \mid w_t \right] + \max_{i \in [m]} \mathbb{E} \left[ [\tilde{\delta}_t^y]_i^2 \mid w_t \right] \leq L^2 \|w_t - w_0\|^2$. Therefore, for $t < T$, the above discussion yields

$$\mathbb{E} \left[ \langle \eta\tilde{\delta}_t, w_t - w_{t+1} \rangle - V_{w_t}(w_{t+1}) \right] \leq \eta^2 \mathbb{E} \left[ \frac{1}{2} \|\tilde{\delta}_t^x\|_2^2 + \max_{i \in [m]} \mathbb{E} \left[ [\tilde{\delta}_t^y]_i^2 \mid w_t \right] \right]$$

$$\leq \eta^2 L^2 \mathbb{E} \|w_t - w_0\|^2 \leq 2\eta^2 L^2 \mathbb{E} V_{w_0}(w_t). \quad (34)$$

To bound the expectation of $\langle \eta\tilde{\Delta}_t, s_t - s_{t+1} \rangle - V_{s_t}(s_{t+1})$ we proceed just as we had with $\tilde{\delta}_t$. By the second CBB property,

$$\|\eta\tilde{\Delta}_t^y\|_\infty = \eta \left\| \tilde{g}_{w_0}^y(w_t) - g^y(w_t) \right\|_\infty \leq 2\eta L + \eta\tau = \frac{2\alpha}{20L} + 1 \leq 1.79,$$

where we used $\eta = \frac{\alpha}{20L^2}$, $\tau = \frac{1}{\eta}$, and $\alpha \leq L$. Therefore, Lemma 7 with $\delta = -\eta\tilde{\Delta}_t^{\mathsf{y}}$ gives

$$\mathbb{E}\left[\langle\eta\tilde{\Delta}_t, s_t - s_{t+1}\rangle - V_{s_t}(s_{t+1})\right] \leq \eta^2\mathbb{E}\sum_{i\in[m]}[s_t^{\mathsf{y}}]_i[\tilde{\Delta}_t^{\mathsf{y}}]_i^2 \leq \eta^2\,\mathbb{E}\max_{i\in[m]}\mathbb{E}\left[[\tilde{\Delta}_t^{\mathsf{y}}]_i^2 \mid w_t\right],$$

where in the final transition we used the fact that $\tilde{\Delta}_t$ conditioned on $w_t$ is independent of $s_t$. Since $\alpha \leq L$, we have $\tau = \frac{20L^2}{\alpha} \geq 20L \geq 2\sqrt{2}L$. Therefore, by Lemma 8, $\mathbb{E}\left[\|\tilde{\Delta}_t^{\mathsf{x}}\|_2^2 \mid w_t\right] + \max_{i\in[m]}\mathbb{E}\left[[\tilde{\Delta}_t^{\mathsf{y}}]_i^2 \mid w_t\right] \leq 2L^2\|w_t - w_0\|^2$. Substituting back, this gives

$$\mathbb{E}\left[\langle\eta\tilde{\Delta}_t, s_t - s_{t+1}\rangle - V_{s_t}(s_{t+1})\right] \leq \eta^2\,\mathbb{E}\left[\tfrac{1}{2}\|\tilde{\Delta}_t^{\mathsf{x}}\|_2^2 + \max_{i\in[m]}\mathbb{E}\left[[\tilde{\Delta}_t^{\mathsf{y}}]_i^2 \mid w_t\right]\right]$$

$$\leq 2\eta^2 L^2\mathbb{E}\|w_t - w_0\|^2 \leq 4\eta^2 L^2\mathbb{E}V_{w_0}(w_t). \qquad (35)$$

Substituting (33), (34) and (35) back into (32), we have

$$\mathbb{E}\max_{u\in\mathcal{Z}}\left\{\frac{1}{T}\sum_{t\in[T]}\langle g(w_t), w_t - u\rangle - \alpha V_{w_0}(u)\right\} \leq \frac{1}{T}\sum_{t\in[T]}\left[10\eta L^2 - \tfrac{\alpha}{2}\right]\mathbb{E}V_{w_0}(w_t) = 0$$

where the last transition follows from $\eta = \frac{\alpha}{20L^2}$; this establishes the bound (11) for the iterates of Algorithm 2 with a CBB gradient estimators. By the argument in the proof of Corollary 1, for $g(z) = (A^\top z^{\mathsf{y}}, -Az^{\mathsf{x}})$, the average of those iterates constitutes an $(\alpha, 0)$-relaxed proximal oracle. $\qquad\square$

### D.5 Complete pseudo-code

---
**Algorithm 4:** Variance reduction for $\ell_2$-$\ell_1$ games

---
**Input**: Matrix $A \in \mathbb{R}^{m\times n}$ with $i$th row $A_{i:}$ and $j$th column $A_{:j}$, target accuracy $\epsilon$
**Output**: A point with expected duality gap below $\epsilon$

1   $L \leftarrow \|A\|_{2\to\infty}$, $\alpha \leftarrow L\sqrt{\frac{n+m}{\mathrm{nnz}(A)}}$, $K \leftarrow \left\lceil\frac{\log(2m)\alpha}{\epsilon}\right\rceil$, $\eta \leftarrow \frac{\alpha}{20L^2}$, $\tau \leftarrow \frac{1}{\eta}$, $T \leftarrow \left\lceil\frac{4}{\eta\alpha}\right\rceil$, $(x_0, y_0) \leftarrow (\mathbf{0}_n, \frac{1}{m}\mathbf{1}_m)$

2 **for** $k = 1, \ldots, K$ **do**

    $\triangleright$ *Relaxed oracle query:*

3     $(x_0, y_0) \leftarrow (z_{k-1}^{\mathsf{x}}, z_{k-1}^{\mathsf{y}})$, $(g_0^{\mathsf{x}}, g_0^{\mathsf{y}}) \leftarrow (A^\top y_0, -Ax_0)$

4     **for** $t = 1, \ldots, T$ **do**

        $\triangleright$ *Gradient estimation:*

5         Sample $i \sim p$ where $p_i = \frac{|[y_{t-1}]_i - [y_0]_i|}{\|y_{t-1} - y_0\|_1}$, sample $j \sim q$ where $q_j = \frac{([x_{t-1}]_j - [x_0]_j)^2}{\|x_{t-1} - x_0\|_2^2}$

6         Set $\tilde{g}_{t-1} = g_0 + \left(A_{i:}\frac{[y_{t-1}]_i - [y_0]_i}{p_i}, -\mathsf{T}_\tau\left(A_{:j}\frac{[x_{t-1}]_j - [x_0]_j}{q_j}\right)\right)$

                                $\triangleright$ $[\mathsf{T}_\tau(v)]_k := \min\{\tau, \max\{-\tau, [v]_k\}\}$

        $\triangleright$ *Mirror descent step:*

7         $x_t \leftarrow \Pi_{\mathcal{X}}\left(\frac{1}{1+\eta\alpha/2}\left(x_{t-1} + \frac{\eta\alpha}{2}x_0 - \eta\tilde{g}_{t-1}^{\mathsf{x}}\right)\right)$         $\triangleright$ $\Pi_{\mathcal{X}}(v) = \frac{v}{\max\{1, \|v\|_2\}}$

8         $y_t \leftarrow \Pi_{\mathcal{Y}}\left(\frac{1}{1+\eta\alpha/2}\left(\log y_{t-1} + \frac{\eta\alpha}{2}\log y_0 - \eta\tilde{g}_{t-1}^{\mathsf{y}}\right)\right)$     $\triangleright$ $\Pi_{\mathcal{Y}}(v) = \frac{e^v}{\|e^v\|_1}$

9     $z_{k-1/2} \leftarrow \frac{1}{T}\sum_{t=1}^{T}(x_t, y_t)$

    $\triangleright$ *Extragradient step:*

10    $z_k^{\mathsf{x}} \leftarrow \Pi_{\mathcal{X}}\left(z_{k-1}^{\mathsf{x}} - \frac{1}{\alpha}A^\top z_{k-1/2}^{\mathsf{y}}\right)$, $z_k^{\mathsf{y}} \leftarrow \Pi_{\mathcal{Y}}\left(\log z_{k-1}^{\mathsf{y}} + \frac{1}{\alpha}Az_{k-1/2}^{\mathsf{x}}\right)$

11 **return** $\frac{1}{K}\sum_{k=1}^{K}z_{k-1/2}$

---

# E  The $\ell_2$-$\ell_2$ setup

**Setup.** In the $\ell_2$-$\ell_2$ setup, both $\mathcal{X} = \mathbb{B}^n$ and $\mathcal{Y} = \mathbb{B}^m$ are Euclidean unit balls, the norm over $\mathcal{Z} = \mathcal{X} \times \mathcal{Y}$ is the Euclidean norm (which is dual to itself), and the distance generating function is $r(z) = \frac{1}{2}\|z\|_2^2$. Under the Euclidean norm, the Lipschitz constant of $g$ is $\|A\|_{2\to 2}$ (the largest singular value of $A$), and we also consider the Frobenius norm $\|A\|_{\mathrm{F}} = (\sum_{i,j} A_{ij}^2)^{1/2}$, i.e. the Euclidean norm of the singular values of $A$.

**Remark 4.** In the $\ell_2$-$\ell_2$ setup, problems of the form $\min_{x\in\mathbb{B}^n} \max_{y\in\mathbb{B}^m} y^\top A x$ are trivial, since the saddle point is always the origin. However, as we explain in Section F.2, our results extend to problems of the form $\min_{x\in\mathbb{B}^n} \max_{y\in\mathbb{B}^m} \{y^\top A x + \phi(x) - \psi(y)\}$ for convex functions $\phi, \psi$, e.g. $\min_{x\in\mathbb{B}^n} \max_{y\in\mathbb{B}^m} \{y^\top A x + b^\top x + c^\top y\}$, which are nontrivial.

Our centered gradient estimator for the $\ell_2$-$\ell_2$ setup is of the form (13), where we sample from

$$p_i(w) = \frac{([w^{\mathsf{y}}]_i - [w_0^{\mathsf{y}}]_i)^2}{\|w^{\mathsf{y}} - w_0^{\mathsf{y}}\|_2^2} \quad \text{and} \quad q_j(w) = \frac{([w^{\mathsf{x}}]_j - [w_0^{\mathsf{x}}]_j)^2}{\|w^{\mathsf{x}} - w_0^{\mathsf{x}}\|_2^2}. \tag{36}$$

The resulting gradient estimator has the explicit form

$$\tilde{g}_{w_0}(w) = g(w_0) + \left( A_{i:} \frac{\|w^{\mathsf{y}} - w_0^{\mathsf{y}}\|_2^2}{[w^{\mathsf{y}} - w_0^{\mathsf{y}}]_i}, -A_{:j} \frac{\|w^{\mathsf{x}} - w_0^{\mathsf{x}}\|_2^2}{[w^{\mathsf{x}} - w_0^{\mathsf{x}}]_j} \right). \tag{37}$$

**Lemma 9.** *In the $\ell_2$-$\ell_2$ setup, the estimator (37) is $(w_0, L)$-centered with $L = \|A\|_{\mathrm{F}}$.*

*Proof.* Unbiasedness follows from the estimator definition. The second property follows from

$$\mathbb{E}\|\tilde{g}_{w_0}(w) - g(w_0)\|_2^2 = \sum_{i\in[m]} \frac{\|A_{i:}\|_2^2}{p_i}([w^{\mathsf{y}}]_i - [w_0^{\mathsf{y}}]_i)^2 + \sum_{j\in[n]} \frac{\|A_{:j}\|_2^2}{q_j}([w^{\mathsf{x}}]_j - [w_0^{\mathsf{x}}]_j)^2$$

$$= \|A\|_{\mathrm{F}}^2 \|w - w_0\|_2^2.$$

$\square$

In Appendix F.3.2 we provide two additional sampling distribution that yield estimators with the same guarantee. We may use these gradient estimator to build an algorithm with a convergence guarantee similar to Theorem 2, except with $\|A\|_{\mathrm{F}}$ instead of $\|A\|_{2\to\infty}$ and 1 instead of $\log(2m)$. This result improves the runtime of Balamurugan and Bach [3] by a $\log(1/\epsilon)$ factor. However, as we discuss in Section 1.3, unlike our $\ell_1$-$\ell_1$ and $\ell_2$-$\ell_1$ results, it is not a strict improvement over the linear-time mirror-prox method, which in the $\ell_2$-$\ell_2$ setting achieves running time $O(\|A\|_{2\to 2} \mathrm{nnz}(A)\epsilon^{-1})$. The regime in which our variance-reduced method has a stronger guarantee than mirror-prox is

$$\mathrm{srank}(A) := \frac{\|A\|_{\mathrm{F}}^2}{\|A\|_{2\to 2}^2} \ll \frac{\mathrm{nnz}(A)}{n+m},$$

i.e. when the spectral sparsity of $A$ is significantly greater than its spatial sparsity.

We remark that $\ell_2$-$\ell_2$ games are closely related to linear regression, as

$$\min_{x\in\mathbb{B}^n} \|Ax - b\|_2^2 = \left( \min_{x\in\mathbb{B}^n} \max_{y\in\mathbb{B}^m} \{y^\top A x - y^\top b\} \right)^2.$$

The smoothness of the objective $\|Ax - b\|_2^2$ is $\|A\|_{2\to 2}^2$, but runtimes of stochastic linear regression solvers typically depend on $\|A\|_{\mathrm{F}}^2$ instead [41, 19, 35, 13, 22, 36, 34, 1]. Viewed in this context, it is not surprising that our $\ell_2$-$\ell_2$ runtime scales as it does.

# F Extensions

In this section we collect a number of results that extend our framework and its applications. In Appendix F.1 we show how to use variance reduction to solve the proximal subproblem to high accuracy. This allows us to implement a relaxed gradient oracle for any monotone operator that admits an appropriate gradient estimator, overcoming a technical limitation in the analysis of Algorithm 2 (see discussion following Corollary 1). In Section F.2 we explain how to extend our results to composite saddle point problems of the form $\min_{x \in \mathcal{X}} \max_{y \in \mathcal{Y}} \{f(x, y) + \phi(x) - \psi(y)\}$, where $f$ admits a centered gradient estimator and $\phi, \psi$ are convex functions. Finally, in Section F.3 we return to the bilinear case and provide a number of alternative gradient estimators for the $\ell_2$-$\ell_1$ and $\ell_2$-$\ell_2$ settings.

## F.1 High precision proximal mappings via variance reduction

Here we describe how to use gradient estimators that satisfy Definition 2 to obtain high precision approximations to the exact proximal mapping, as well as a relaxed proximal oracle valid beyond the bilinear case. Algorithm 5 is a modification of Algorithm 2, where we restart the mirror-descent iteration $N$ times, with each restarting constituting a *phase*. In each phase, we re-center the gradient estimator $g$, but regularize towards the original initial point $w_0$. To analyze the performance of the algorithm, we require two properties of proximal mappings with general Bregman divergences (10).

**Lemma 10.** *Let $g$ by a monotone operator, let $z \in \mathcal{Z}$ and let $\alpha > 0$. Then, for every $w \in \mathcal{Z}$, $z_\alpha = \mathrm{Prox}_z^\alpha(g)$ satisfies*

$$\langle g(w) + \alpha \nabla V_z(w), w - z_\alpha \rangle \geq \alpha V_{z_\alpha}(w) + \alpha V_w(z_\alpha).$$

*Proof.* By definition of $z_\alpha$, $\langle g(z_\alpha) + \alpha \nabla V_z(z_\alpha), z_\alpha - w \rangle \leq 0$ for all $w \in \mathcal{Z}$. Therefore

$$\begin{aligned}
\langle g(w) + \alpha \nabla V_z(w), w - z_\alpha \rangle &\geq \langle g(w) + \alpha \nabla V_z(w), w - z_\alpha \rangle + \langle g(z_\alpha) + \alpha \nabla V_z(z_\alpha), z_\alpha - w \rangle \\
&= \langle g(w) - g(z_\alpha), w - z_\alpha \rangle + \alpha \langle \nabla V_z(w) - \nabla V_z(z_\alpha), w - z_\alpha \rangle \\
&\underset{(i)}{\geq} \alpha \langle \nabla V_z(w) - \nabla V_z(z_\alpha), w - z_\alpha \rangle \underset{(ii)}{=} \alpha V_{z_\alpha}(w) + \alpha V_w(z_\alpha),
\end{aligned}$$

where $(i)$ follows from monotonicity of $g$ and $(ii)$ holds by definition of the Bregman divergence. $\qquad\square$

**Lemma 11.** *Let $g$ be a monotone operator and let $\alpha > 0$. Then, for every $z \in \mathcal{Z}$, $z_\alpha = \mathrm{Prox}_z^\alpha(g)$ satisfies*

$$V_{z_\alpha}(z) + V_z(z_\alpha) \leq \frac{\|g(z)\|_* \|z - z_\alpha\|}{\alpha} \leq \frac{\|g(z)\|_*^2}{\alpha^2}.$$

*Proof.* Using Lemma 10 with $w = z$ gives

$$\alpha V_{z_\alpha}(z) + \alpha V_z(z_\alpha) \leq \langle g(z) + \alpha \nabla V_z(z), z - z_\alpha \rangle \leq \langle g(z), z - z_\alpha \rangle,$$

where we used the fact that $z$ minimizes the convex function $V_z(\cdot)$ and therefore $\langle \nabla V_z(z), z - u \rangle \leq 0$ for all $u \in \mathcal{Z}$. Writing $\langle g(z), z - z_\alpha \rangle \leq \|g(z)\|_* \|z - z_\alpha\|$ gives the first bound in the lemma. Next, strong convexity of $r$ implies

$$\|z - z_\alpha\|^2 \leq V_{z_\alpha}(z) + V_z(z_\alpha) \leq \frac{\|g(z)\|_* \|z - z_\alpha\|}{\alpha},$$

and the second bound follows from dividing by $\|z - z_\alpha\|$. $\qquad\square$

We now state the main convergence result for Algorithm 5.

**Proposition 4.** *Let $\alpha, L > 0$, let $w_0 \in \mathcal{Z}$, let $\tilde{g}_z$ be $(z, L)$-centered for monotone $g$ and every $z \in \mathcal{Z}$ and let $z_\alpha = \mathrm{Prox}_{w_0}^\alpha(g)$. Then, for $\eta = \frac{\alpha}{8L^2}$, $T \geq \frac{4}{\eta\alpha} = \frac{32L^2}{\alpha^2}$, and any $N \in \mathbb{N}$ the output $\hat{w}_N$ of Algorithm 5 satisfies*

$$\mathbb{E} V_{\hat{w}_N}(z_\alpha) \leq 2^{-N} V_{w_0}(z_\alpha). \tag{38}$$

**Algorithm 5:** `RestartedInnerLoop`$(w_0, z \mapsto \tilde{g}_z, \alpha)$

---

**Input**: Initial $w_0 \in \mathcal{Z}$, centered gradient estimator $\tilde{g}_z \; \forall z \in \mathcal{Z}$, oracle quality $\alpha > 0$
**Parameters**: Step size $\eta$, inner iteration count $T$, phase count $N$
**Output**: Point $\hat{w}_N$ satisfying $\mathbb{E}V_{\hat{w}_N}(z_\alpha) \le 2^{-N} V_{w_0}(z_\alpha)$ where $z_\alpha = \text{Prox}^\alpha_{w_0}(g)$ (for appropriate $\tilde{g}$, $\eta$, $T$)

1  Set $\hat{w}_0 \leftarrow w_0$
2  **for** $n = 1, \ldots, N$ **do**
3  $\quad$ Prepare centered gradient estimator $\tilde{g}_{\hat{w}_{n-1}}$ $\qquad\qquad$ ▷ e.g. by computing $g(\hat{w}_{n-1})$
4  $\quad$ Draw $\hat{T}$ uniformly from $[T]$
5  $\quad$ $w_0^{(n)} \leftarrow \hat{w}_{n-1}$
6  $\quad$ **for** $t = 1, \ldots, \hat{T}$ **do**
7  $\quad\quad$ $\left\lfloor \; w_t^{(n)} \leftarrow \arg\min_{w \in \mathcal{Z}} \left\{ \left\langle \tilde{g}_{\hat{w}_{n-1}}\big(w_{t-1}^{(n)}\big), w \right\rangle + \alpha V_{w_0}(w) + \frac{1}{\eta} V_{w_{t-1}^{(n)}}(w) \right\} \right.$
8  $\quad$ $\hat{w}_n \leftarrow w_{\hat{T}}^{(n)}$
9  **return** $\hat{w}_N$

---

*Proof.* Fix a phase $n \in [N]$. For every $u \in \mathcal{Z}$ we have the mirror descent regret bound

$$\sum_{t \in [T]} \left\langle \tilde{g}_{\hat{w}_{n-1}}\big(w_t^{(n)}\big) + \alpha \nabla V_{w_0}\big(w_t^{(n)}\big), w_t^{(n)} - u \right\rangle \le \frac{V_{\hat{w}_{n-1}}(u)}{\eta} + \frac{\eta}{2} \sum_{t \in [T]} \left\| \tilde{g}_{\hat{w}_{n-1}}\big(w_t^{(n)}\big) - g(\hat{w}_{n-1}) \right\|_*^2 ;$$

see Lemma 4 in Appendix A.2, with $Q(z) = \eta \langle g(\hat{w}_{n-1}), z \rangle + \eta \alpha V_{w_0}(z)$. Choosing $u = z_\alpha$, taking expectation and using Definition 2 gives

$$\mathbb{E} \sum_{t \in [T]} \left\langle g\big(w_t^{(n)}\big) + \alpha \nabla V_{w_0}\big(w_t^{(n)}\big), w_t^{(n)} - z_\alpha \right\rangle \le \frac{\mathbb{E}V_{\hat{w}_{n-1}}(z_\alpha)}{\eta} + \frac{\eta L^2}{2} \sum_{t \in [T]} \mathbb{E} \left\| w_t^{(n)} - \hat{w}_{n-1} \right\|^2 . \tag{39}$$

(Note that $z_\alpha$ is a function of $w_0$ and hence independent of stochastic gradient estimates.) By the triangle inequality and strong convexity of $r$,

$$\|w_t^{(n)} - \hat{w}_{n-1}\|^2 \le 2\|z_\alpha - \hat{w}_{n-1}\|^2 + 2\|w_t^{(n)} - z_\alpha\|^2 \le 4V_{\hat{w}_{n-1}}(z_\alpha) + 4V_{z_\alpha}\big(w_t^{(n)}\big). \tag{40}$$

By Lemma 10 we have that for every $t \in [T]$

$$\left\langle g\big(w_t^{(n)}\big) + \alpha \nabla V_{w_0}\big(w_t^{(n)}\big), w_t^{(n)} - z_\alpha \right\rangle \ge \alpha V_{w_t^{(n)}}(z_\alpha) + \alpha V_{z_\alpha}\big(w_t^{(n)}\big). \tag{41}$$

Substituting the bounds (40) and (41) into the expected regret bound (39) and rearranging gives

$$\frac{1}{T} \sum_{t \in [T]} \mathbb{E}V_{w_t^{(n)}}(z_\alpha) \le \left( \frac{1}{\eta \alpha T} + \frac{2\eta L^2}{\alpha} \right) \mathbb{E}V_{\hat{w}_{n-1}}(z_\alpha) + \frac{2\eta L^2 - \alpha}{\alpha T} \sum_{t \in [T]} \mathbb{E}V_{z_\alpha}\big(w_t^{(n)}\big) \le \frac{1}{2} \mathbb{E}V_{w_t^{(n-1)}}(z_\alpha),$$

where in the last transition we substituted $\eta = \frac{\alpha}{8L^2}$ and $T \ge \frac{4}{\eta \alpha}$. Noting that $\frac{1}{T} \sum_{t \in [T]} \mathbb{E}V_{w_t^{(n)}}(z_\alpha) = \mathbb{E}V_{\hat{w}_n}(z_\alpha)$ and recursing on $n$ completes the proof. $\qquad\square$

The linear convergence bound (38) combined with Lemma 11 implies that Algorithm 5 implements a relaxed proximal oracle.

**Corollary 2.** *Let $G, D > 0$ be such that $\|g(z)\|_* \le G$ and $\|z - z'\| \le D$ for every $z, z' \in \mathcal{Z}$ and let $\varepsilon > 0$. Then, in the setting of Proposition 4 with $N \ge 1 + 2\log_2\left( \frac{G(G+2LD)}{\alpha \varepsilon} \right)$, we have that $\mathcal{O}(w_0) = \texttt{RestartedInnerLoop}(w_0, \tilde{g}, \alpha)$ is an $(\alpha, \varepsilon)$-relaxed proximal oracle.*

*Proof.* Let $\hat{w} = \texttt{RestartedInnerLoop}(w_0, \tilde{g}, \alpha)$ and let $z_\alpha = \text{Prox}^\alpha_{w_0}(g)$. For every $u \in \mathcal{Z}$, we have

$$\langle g(\hat{w}), \hat{w} - u \rangle = \langle g(z_\alpha), z_\alpha - u \rangle + \langle g(z_\alpha), \hat{w} - z_\alpha \rangle + \langle g(\hat{w}) - g(z_\alpha), \hat{w} - u \rangle .$$

By the definition (10) of $z_\alpha$ we have $\langle g(z_\alpha), z_\alpha - u\rangle \le \alpha V_{w_0}(u)$. By Hölder's inequality and the assumption that $g$ is bounded, we have $\langle g(z_\alpha), \hat{w} - z_\alpha\rangle \le G\|\hat{w} - z_\alpha\|$. Finally, since $g$ is $2L$-Lipschitz (see Remark 1) and $\|\hat{w} - u\| \le D$ by assumption, we have $\langle g(\hat{w}) - g(z_\alpha), \hat{w} - u\rangle \le 2LD\|\hat{w} - z_\alpha\|$. Substituting back these three bounds and rearranging yields

$$\langle g(\hat{w}), \hat{w} - u\rangle - \alpha V_{w_0}(u) \le (G + 2LD)\|\hat{w} - z_\alpha\| \le (G + 2LD)\sqrt{2V_{\hat{w}}(z_\alpha)},$$

where the last bound is due to strong convexity of $r$. Maximizing over $u$ and taking expectation, we have by Jensen's inequality and Proposition 4,

$$\mathbb{E}\max_{u\in\mathcal{Z}}\{\langle g(\hat{w}), \hat{w} - u\rangle - \alpha V_{w_0}(u)\} \le (G+2LD)\sqrt{2\mathbb{E}V_{\hat{w}}(z_\alpha)} \le 2^{-(N-1)/2}(G+2LD)\sqrt{V_{w_0}(z_\alpha)}.$$

Lemma 11 gives us $\sqrt{V_{w_0}(z_\alpha)} \le \sqrt{\|g(w_0)\|_*^2/\alpha^2} \le G/\alpha$, and therefore $N \ge 1 + 2\log_2\left(\frac{G(G+2LD)}{\alpha\varepsilon}\right)$ establishes the oracle property $\mathbb{E}\max_{u\in\mathcal{Z}}\{\langle g(\hat{w}), \hat{w} - u\rangle - \alpha V_{w_0}(u)\} \le \varepsilon$. $\qquad\square$

**Remark 5.** In the $\ell_2$-$\ell_1$ setup of Section 4.2, Proposition 4 and Corollary 2 extend straightforwardly to centered-bounded-biased gradient estimators (Definition 3) using arguments from the proof of Proposition 3.

Since Algorithm 5 computes a highly accurate approximation of the proximal mapping, it is reasonable to expect that directly iterating $z_k = \texttt{RestartedInnerLoop}(z_{k-1}, \tilde{g}, \alpha)$ for $k \in [K]$ would yield an $O(\alpha\Theta/K)$ error bound, without requiring the extragradient step in Algorithm 1. However, we could not show such a bound without additionally requiring uniform smoothness of the distance generating function $r$, which does not hold for the negative entropy we use in the $\ell_1$ setting.

### F.2 Composite saddle point problems

Consider the "composite" saddle point problem of the form

$$\min_{x\in\mathcal{X}}\max_{y\in\mathcal{Y}}\{f(x, y) + \phi(x) - \psi(y)\},$$

where $\nabla f$ admits a centered gradient estimator and $\phi, \psi$ are "simple" convex functions in the sense they have efficiently-computable proximal mappings. As usual in convex optimization, it is straightforward to extend our framework to this setting. Let $\Upsilon(z) := \phi(z^\mathsf{x}) + \psi(z^\mathsf{y})$ so that $g(z) + \nabla\Upsilon(z)$ denotes the (sub-)gradient mapping for the composite problem at point $z$. Algorithmically, the extension consists of changing Line 4 of Algorithm 1 to

$$z_k \leftarrow \arg\min_{z\in\mathcal{Z}}\left\{\langle g(z_{k-1/2}) + \nabla\Upsilon(z_{k-1/2}), z\rangle + \alpha V_{z_{k-1}}(z)\right\},$$

changing line 2 of Algorithm 2 to

$$w_t \leftarrow \arg\min_{w\in\mathcal{Z}}\left\{\langle \tilde{g}_{w_0}(w_{t-1}), w\rangle + \Upsilon(w) + \frac{\alpha}{2}V_{w_0}(w) + \frac{1}{\eta}V_{w_{t-1}}(w)\right\},$$

and similarly adding $\Upsilon(w)$ to the minimization in line 7 of Algorithm 5.

Analytically, we replace $g$ with $g + \nabla\Upsilon$ in the duality gap bound (8), Definition 1 (relaxed proximal oracle), and Proposition 1 and its proof, which holds without further change. To implement the composite relaxed proximal oracle we still assume a centered gradient estimator for $g$ only. However, with the algorithmic modifications described above, the guarantee (11) of Proposition 2 now has $g + \nabla\Upsilon$ instead of $g$; the only change to the proof is that we now invoke Lemma 4 (in Appendix A.2) with the composite term $\eta\left[\langle g(w_0), z\rangle + \Upsilon(z) + \frac{\alpha}{2}V_{w_0}(z)\right]$, and the bound (20) becomes

$$\sum_{t\in[T]}\langle \tilde{g}_{w_0}(w_t) + \nabla\Upsilon(w_t) + \frac{\alpha}{2}\nabla V_{w_0}(w_t), w_t - u\rangle \le \frac{V_{w_0}(u)}{\eta} + \frac{\eta}{2}\sum_{t\in[T]}\|\tilde{\delta}_t\|_*^2.$$

Proposition 3, Proposition 4 and Corollary 2 similarly extend to the composite setup.

The only point in our development that does not immediately extend to the composite setting is Corollary 1 and its subsequent discussion. There, we argue that Algorithm 2 implements a relaxed proximal oracle only when $\langle g(z), z - u\rangle$ is convex in $z$ for all $u$, which is the case for bilinear $f$. However, this condition might fail for $g + \nabla\Upsilon$ even when it holds for $g$. In this case, we may still use the oracle implementation guaranteed by Corollary 2 for any convex $\Upsilon$.

## F.3 Additional gradient estimators

We revisit the bilinear setting studied in Section 4 and provide additional gradient estimators that meet our variance requirements. In Section F.3.1 we consider $\ell_2$-$\ell_1$ games and construct an "oblivious" estimator for the $\mathcal{Y}$ component of the gradient that involves sampling from a distribution independent of the query point. In Section F.3.2 we describe two additional centered gradient estimators for $\ell_2$-$\ell_2$ games; one of them is the "factored splits" estimator proposed in [3].

### F.3.1 $\ell_2$-$\ell_1$ games

Consider the $\ell_2$-$\ell_1$ setup described in the beginning of Section 4.2. We describe an alternative for the $\mathcal{Y}$ component of (29), that is "oblivious" in the sense that it involves sampling from distributions that do not depend on the current iterate. The estimator generates each coordinate of $\tilde{g}_{w_0}^{\mathsf{y}}$ independently in the following way: for every $i \in [m]$ we define the probability $q^{(i)} \in \Delta^n$ by

$$q_j^{(i)} = A_{ij}^2 / \|A_{i:}\|_2^2, \ \ \forall j \in [n].$$

Then, independently for every $i \in [m]$, draw $j(i) \sim q^{(i)}$ and set

$$[\tilde{g}_{w_0}^{\mathsf{y}}(w)]_i = -[Aw_0^{\mathsf{x}}]_i - \mathsf{T}_\tau \left( A_{ij(i)} \frac{[w^{\mathsf{x}}]_{j(i)} - [w_0^{\mathsf{x}}]_{j(i)}}{q_{j(i)}^{(i)}} \right), \tag{42}$$

where $\mathsf{T}_\tau$ is the clipping operator defined in (29). Note that despite requiring $m$ independent samples from different distributions over $n$ elements, $\tilde{g}_{w_0}^{\mathsf{y}}$ still admits efficient evaluation. This is because the distributions $q^{(i)}$ are fixed in advance, and we can pre-process them to perform each of the $m$ samples in time $O(1)$ [42]. However, the oblivious gradient estimator produces fully dense estimates regardless of the sparsity of $A$, which limits its running time guarantees to terms proportional to $m$ rather than the maximum number of nonzero elements in columns of $A$.

The oblivious estimator has the same "centered-bounded-biased" properties (Definition 3) as the "dynamic" estimator (29).

**Lemma 12.** *In the $\ell_2$-$\ell_1$ setup, a gradient estimator with $\mathcal{X}$ block as in (29) and $\mathcal{Y}$ block as in (42) is $(w_0, L, \tau)$-CBB with $L = \|A\|_{2\to\infty}$.*

*Proof.* We show the bias bound similarly to the proof of Lemma 6,

$$\left| \mathbb{E} \left[ \tilde{g}_{w_0}^{\mathsf{y}}(w) - g^{\mathsf{y}}(w) \right]_i \right| \leq \sum_{j \in \mathcal{J}_\tau(i)} |A_{ij}| \, |[w^{\mathsf{x}}]_j - [w_0^{\mathsf{x}}]_j|$$

for all $i \in [m]$, where

$$\mathcal{J}_\tau(i) = \left\{ j \in [n] \mid \mathsf{T}_\tau \left( \frac{A_{ij}}{q_j^{(i)}} ([w^{\mathsf{x}}]_j - [w_0^{\mathsf{x}}]_j) \right) \neq \frac{A_{ij}}{q_j^{(i)}} ([w^{\mathsf{x}}]_j - [w_0^{\mathsf{x}}]_j) \right\}.$$

Note that $j \in \mathcal{J}_\tau(i)$ if and only if

$$\left| \frac{A_{ij}}{q_j^{(i)}} ([w^{\mathsf{x}}]_j - [w_0^{\mathsf{x}}]_j) \right| = \frac{\|A_{i:}\|_2^2 \, |[w^{\mathsf{x}}]_j - [w_0^{\mathsf{x}}]_j|}{|A_{ij}|} > \tau \Rightarrow |A_{ij}| \leq \frac{1}{\tau} \|A_{i:}\|_2^2 \, |[w^{\mathsf{x}}]_j - [w_0^{\mathsf{x}}]_j| \, .$$

Therefore,

$$\sum_{j \in \mathcal{J}_\tau(i)} |A_{ij}| \, |[w^{\mathsf{x}}]_j - [w_0^{\mathsf{x}}]_j| \leq \frac{1}{\tau} \|A_{i:}\|_2^2 \sum_{j \in \mathcal{J}_\tau} |[w^{\mathsf{x}}]_j - [w_0^{\mathsf{x}}]_j|^2 = \frac{1}{\tau} \|A_{i:}\|_2^2 \, \|w^{\mathsf{x}} - w_0^{\mathsf{x}}\|_2^2$$

and $\left\| \mathbb{E} \tilde{g}_{w_0}^{\mathsf{y}}(w) - g^{\mathsf{y}}(w) \right\|_\infty \leq \frac{L^2}{\tau} \|w^{\mathsf{x}} - w_0^{\mathsf{x}}\|_2^2$ follows by taking the maximum over $i \in [m]$.

The second property follows exactly as in the proof of Lemma 6. For the third property, note that the bound (26) on the $\mathcal{X}$ component still holds, and that for each $i \in [m]$ we have $q_j^{(i)} = A_{ij}^2 / \|A_{i:}\|_2^2$ and

$$
\begin{aligned}
\mathbb{E}\left[\tilde{g}_{w_0}^{\mathsf{y}}(w) - g^{\mathsf{y}}(w)\right]_i^2 &= \sum_{j \in [n]} q_j^{(i)} \left(\mathsf{T}_\tau\left(\frac{A_{ij}}{q_j^{(i)}}([w^{\mathsf{x}}]_j - [w_0^{\mathsf{x}}]_j)\right)\right)^2 \\
&\leq \sum_{j \in [n]} q_j^{(i)} \left(\frac{A_{ij}}{q_j^{(i)}}([w^{\mathsf{x}}]_j - [w_0^{\mathsf{x}}]_j)\right)^2 = \|A_{i:}\|_2^2 \|w^{\mathsf{x}} - w_0^{\mathsf{x}}\|_2^2.
\end{aligned}
$$

$\square$

### F.3.2 $\ell_2$-$\ell_2$ games

In the $\ell_2$-$\ell_2$ setup described in Section E it is possible to use a completely oblivious gradient estimator. It has the form (13) with the following sampling distributions that do not depend on $w_0, w$,

$$
p_i = \frac{\|A_{i:}\|_2^2}{\|A\|_{\mathrm{F}}^2} \quad \text{and} \quad q_j = \frac{\|A_{:j}\|_2^2}{\|A\|_{\mathrm{F}}^2}. \tag{43}
$$

Balamurugan and Bach [3] use these sampling distributions, referring to them as "factored splits." Another option is to use the dynamic sampling probabilities

$$
p_i(w) = \frac{\|A_{i:}\|_2 \, |[w^{\mathsf{y}}]_i - [w_0^{\mathsf{y}}]_i|}{\sum_{i' \in [m]} \|A_{i':}\|_2 \, |[w^{\mathsf{y}}]_{i'} - [w_0^{\mathsf{y}}]_{i'}|} \quad \text{and} \quad q_j(w) = \frac{\|A_{:j}\|_2 \, |[w^{\mathsf{x}}]_j - [w_0^{\mathsf{x}}]_j|}{\sum_{j' \in [n]} \|A_{:j'}\|_2 \, |[w^{\mathsf{x}}]_{j'} - [w_0^{\mathsf{x}}]_{j'}|}. \tag{44}
$$

Both the distributions above yield centered gradient estimators.

**Lemma 13.** *In the $\ell_2$-$\ell_2$ setup, the estimator* (13) *with either sampling probabilities* (43) *or* (44) *is* $(w_0, L)$-*centered for* $L = \|A\|_{\mathrm{F}}$.

*Proof.* Unbiasedness follows from the estimator definition. For the oblivious sampling strategy (43) the second property follows from

$$
\begin{aligned}
\mathbb{E}\|\tilde{g}_{w_0}(w) - g(w_0)\|_2^2 &= \sum_{i \in [m]} \frac{\|A_{i:}\|_2^2}{p_i}([w^{\mathsf{y}}]_i - [w_0^{\mathsf{y}}]_i)^2 + \sum_{j \in [n]} \frac{\|A_{:j}\|_2^2}{q_j}([w^{\mathsf{x}}]_j - [w_0^{\mathsf{x}}]_j)^2 \\
&= \|A\|_{\mathrm{F}}^2 \|w - w_0\|_2^2.
\end{aligned}
$$

For the dynamic sampling strategy (44), we have

$$
\begin{aligned}
\mathbb{E}\|\tilde{g}_{w_0}(w) - g(w_0)\|_2^2 &= \left(\sum_{i' \in [m]} \|A_{i':}\|_2 \, |[w^{\mathsf{y}}]_{i'} - [w_0^{\mathsf{y}}]_{i'}|\right)^2 + \left(\sum_{j' \in [n]} \|A_{:j'}\|_2 \, |[w^{\mathsf{x}}]_{j'} - [w_0^{\mathsf{x}}]_{j'}|\right)^2 \\
&\leq \|A\|_{\mathrm{F}}^2 \|w - w_0\|_2^2,
\end{aligned}
$$

where the inequality is due to Cauchy–Schwarz. $\square$

We remark that out of the three sampling strategies (36), (43) and (44), only for (44) the bound $\mathbb{E}\|\tilde{g}_{w_0}(w) - g(w_0)\|_2^2 \leq \|A\|_{\mathrm{F}}^2 \|w - w_0\|_2^2$ is an inequality, whereas for the other two it holds with equality. Consequently, the dynamic sampling probabilities (44) might be preferable in certain cases.