[Reviews · NeurIPS 2019]

Reviewer 1



In this paper, the authors are interested in the problem of bilinear minimax games. By using tools from the optimization techniques of variance reduction, the authors show how to attain an eps-optimal (in additive error) solution to the problem in total time nnz(A) + sqrt{nnz(A)*n}/eps. Furthermore, their results hold for both l_1 - l_1 and l_1 - l_2 games. One of the key technical contributions is an approach called “sampling from the difference”, which leads to a desired variance bound. Various results in computational geometry, such as maximum inscribed ball and minimum enclosing ball, can also be recovered from these results. ====== Strengths ====== The authors present a variance reduced method for solving the bilinear saddle point problem in time nnz(A) + sqrt{nnz(A)*n}/eps, thus improving upon Nemirovski’s mirror-prox method by a factor of sqrt{nnz(A)/n}. The paper is well-written with a clear focus and detailed accounting of how the newly proposed method compares to previous work. Although the new method does not provide unilateral improvements, the authors carefully lay out the range of parameters in which their method is better, as well as when the other methods dominate, thus further illustrating the power of variance reduction for bilinear saddle point games. ====== Weaknesses ====== The authors are missing some recent related works concerning bilinear saddle point games. In particular, Sherman [3] has established rates for bilinear saddle point games that improve upon the extragradient methods of Nemirovski [1] and Nesterov [2], by combining a special type of regularizer with the dual extrapolation method in [2]. Concerning dependence on the “range” of the regularizers, it would be recommended that the authors provide additional discussion comparing these methods (and perhaps mention subsequent work by Sidford and Tian [4], which focuses on the special case of l_infty regression). Along the same lines of the “range” considerations, and though I may be mistaken, is there a missing diameter/distance term for the runtime of the extragradient methods in eq.(1)? While I know it is the case that the log(m) and log(n) factors are being hidden in \tilde{O} (which is fine for l_1 - l_1 games), for l_1 - l_2 games, shouldn’t it be necessary to account for the distance from a center point (w.r.t. the distance generating function), which may introduce an extra poly(m) term? At the very least, it would be helpful to clarify the dependence on the distance generating function(s) (including for mirror-prox), and their relation to the various min/max domains. ============ I am generally inclined to accept this paper, as the work provides an interesting insight into faster convergence rates for matrix games (in certain regimes). [1] Nemirovski, A. "Prox-method with rate of convergence O (1/t) for variational inequalities with Lipschitz continuous monotone operators and smooth convex-concave saddle point problems." SIAM Journal on Optimization 15, no. 1 (2004): 229-251. [2] Nesterov, Y. "Dual extrapolation and its applications to solving variational inequalities and related problems." Mathematical Programming 109, no. 2-3 (2007): 319-344. [3] Sherman, J. "Area-convexity, l∞ regularization, and undirected multicommodity flow." In Proceedings of the 49th Annual ACM SIGACT Symposium on Theory of Computing, pp. 452-460. ACM, 2017. [4] Sidford, A., and Tian, K. "Coordinate methods for accelerating ℓ∞ regression and faster approximate maximum flow." In 2018 IEEE 59th Annual Symposium on Foundations of Computer Science (FOCS), pp. 922-933. IEEE, 2018. ============================ After reading the authors' response, I appreciate the clarification and acknowledgement of the additional related work, and I have adjusted my score accordingly. ============================

Reviewer 2



######################################################## Having read the author's response, I maintain my comments and score. The added comparison with previous work will be a useful addition to the paper. ######################################################## This paper was a very nice read. The ideas were clearly presented, the proofs were communicated both at a high, intuitive level and in detail in the appendix, and the algorithms are a substantial improvement over previous work in certain regimes. The paper is, to my knowledge, original and the technical details are correct. In particular, the formalization in Definition 2 is quite interesting, and leads to interesting gradient estimators which, unlike in typical variance reduction methods, are constructed in a different way depending on the query point. Overall, this paper clearly presents an interesting and seemingly quite practical algorithm for solving an important class of optimization problems.

Reviewer 3



This is a very original revisitation of the classic mirror-prox framework of Nemirovski. It allows the application of variance-reduced sgd to a new setting, minmax optimization. The variance reduction technique of sampling from the difference may be of independent interest. The paper is well-written.

[Author Response · NeurIPS 2019]

We thank the reviewers for their time and attention. Below, we address each review in turn. In addition, we describe a subtle technical correction we made to our paper shortly after submission, which does not change our runtime guarantees.

**Reviewer 1** Thank you for your review; we are glad you found our paper interesting and well-written. Below, we address in detail the two concerns raised in the review, starting with a comparison to Sherman (2017). We hope this comparison meets your requirement for raising our paper's score.

*Comparison to algorithms for $\ell_\infty$-$\ell_1$ games.* The recent papers by Sherman (2017) and Sidford and Tian (2018) consider a different setting than we do, and their developments do not imply runtime improvements for our setting. Specifically, these papers consider bilinear saddle-point problems $\min_{x \in \mathcal{X}} \max_{y \in \mathcal{Y}} y^\top A x$ where the domain $\mathcal{X}$ is the box ($\ell_\infty$ ball) while $\mathcal{Y}$ is the simplex. As Sherman explains in his introduction, the $\ell_\infty$ domain is challenging because no distance generating function has both 1-strong-convexity w.r.t. $\ell_\infty$ and range sublinear in dimension. Sherman's development side-steps this challenge using finer-grained notions of convexity and Nesterov's dual extrapolation method to obtain improved runtime guarantees for $\ell_\infty$-$\ell_1$ games. Sidford and Tian attack this challenge using local notions of smoothness and a randomized coordinate method, and obtain improved runtimes for column-sparse $A$.

In contrast, this challenge does not exist in the $\ell_1$-$\ell_1$ and $\ell_1$-$\ell_2$ games that we study, because in these settings suitable distance generating functions are readily available: negative entropy for $\ell_1$ and Euclidean norm for $\ell_2$. Consequently, the ideas in Sherman (2017) and Sidford and Tian (2018) do not imply runtime improvements for our settings. These works do not consider variance reduction, on which our paper crucially relies. In future work, we intend to explore whether our variance reduction techniques can provide benefits in the $\ell_\infty$-$\ell_1$ setting. When revising our paper we will be sure to cite Sherman (2017) and Sidford and Tian (2018) and compare them to our development—thank you for pointing out the importance of this comparison.

*The range of the distance generating function (dgf).* For $\ell_1$-$\ell_2$ games the range of our dgf is $\frac{1}{2} + \log m$. More generally, in our paper the range $\Theta$ does not introduce polynomial dependence on problem dimension; the runtime bounds in Theorems 1 and 2 account for $\Theta$ and also include logarithmic terms. For the $n$-dimensional simplex ($\ell_1$ domain) we use negative entropy as the dgf, and it has range $\Theta = \log n$. For the unit Euclidean ball ($\ell_2$ domain), our dgf is half the Euclidean norm, so that $\Theta = 1/2$, regardless of the dimension. (Note that, as is standard in the literature, we consider simplices and Euclidean balls of unit norm. This is without loss of generality as scaling of the domain is equivalent to scaling of the matrix $A$, and we account for its norm via the parameter $L$.) In Eqs. (1) and (2) in the introduction we substituted the relevant values of $\Theta$ into the runtime guarantees. However, since this creates confusion, we will revise the introduction to clarify the contribution of the range $\Theta$ to the runtime bound.

**Reviewer 2** Thank you for the kind review; we are pleased that our development came across clearly. We hope that our repackaging of Nemirovski's ideas and our "sampling from the difference" technique will inspire and assist future improvements in minimax optimization and variational inequalities.

**Reviewer 3** Thank you for the generous review and for recognizing the novelty and significance of our results. Indeed, extending the regime in which stochastic (and possibly variance-reduced) methods aid minimax game solution is an exciting direction for further research, in which we are currently engaged.

**Averaged vs. random iterates** In convex optimization, returning a random iterate and returning the average of the iterates often result in equivalent guarantees. However, this is not the case in our paper, and we must do the latter. Therefore, we changed Algorithm 1 (OuterLoop) to run for the full $K$ iterations (no random stopping) and return the average $\bar{z}_K = \frac{1}{K} \sum_{k=1}^{K} z_{k-1/2}$. This way, the proof of Lemma 1 implies a bound on the duality gap at $\bar{z}_K$ as defined in line 154, via standard convexity arguments (cf. [23] page 8); we do not get this guarantee with a random iterate.

Similarly, we changed Algorithm 2 (InnerLoop) to run for $T$ iterations and return $\bar{w}_T = \frac{1}{T} \sum_{t=1}^{T} w_t$. This way, for bilinear games $\bar{w}_T$ satisfies the $\alpha$-proximal oracle property. To see this, note that when $g(w) = (A^\top w^y, -A w^x)$ we have $\frac{1}{T} \sum_{t=1}^{T} \langle g(w_t), w_t - u \rangle = \langle g(\bar{w}_T), \bar{w}_T - u \rangle$ and therefore the $\alpha$-proximal oracle property holds by the bound in Lemma 2; we do not get this property with a random iterate. In the revised paper we also include a proximal oracle implementation valid for general games (i.e. not only bilinear), that returns the last iterate (i.e. does not perform averaging) but requires a number of restarts logarithmic in $1/\epsilon$.

We note that the proofs of Lemmas 1 and 2 in the submitted paper are already fully compatible with these algorithmic modifications, and that our main results (Theorem 1, Corollary 1 and Theorem 2) hold unchanged.

[Meta-Review · NeurIPS 2019]

The paper gives an algorithm for solving minimax matrix games faster, that improve on existing methods in high accuracy/sparse regimes. The new approach is based on an extension of Nemirovsky's mirror-prox algorithm with a novel variance-reduced approximate gradient estimator, which the reviewers found to be significant and of independent interest. The paper contains strong contributions and techniques, and given the high praise of the reviewers, it is a clear accept.